# Varying demands for cognitive control reveals shared neural processes supporting semantic and episodic memory retrieval

Deniz Vatansever [1,2✉], Jonathan Smallwood [2] & Elizabeth Jefferies[2]

The categorisation of long-term memory into semantic and episodic systems has been an influential catalyst for research on human memory organisation. However, the impact of variable cognitive control demands on this classical distinction remains to be elucidated. Across two independent experiments, here we directly compare neural processes for the controlled versus automatic retrieval of semantic and episodic memory. In a multi-session functional magnetic resonance imaging experiment, we first identify a common cluster of cortical activity centred on the left inferior frontal gyrus and anterior insular cortex for the retrieval of both weakly-associated semantic and weakly-encoded episodic memory traces. In an independent large-scale individual difference study, we further reveal a common neural circuitry in which reduced functional interaction between the identified cluster and ventromedial prefrontal cortex, a default mode network hub, is linked to better performance across both memory types. Our results provide evidence for shared neural processes supporting the controlled retrieval of information from functionally distinct long-term memory systems.

[1] Institute of Science and Technology for Brain-inspired Intelligence, Fudan University, Shanghai, PR China. [2] Department of Psychology, University of York, York, UK. ✉email: deniz@fudan.edu.cn

Our ability to understand and predict the world around us hinges upon our long-term memory stores that are historically divided into two distinct systems[1]. While the semantic memory system provides a conceptual framework that describes similarities in meaning when words and objects are encountered under variable contexts (e.g., bees are flying insects with yellow and black stripy colours that produce honey), the episodic memory system encodes our personal experiences characterised by the co-occurrence of words and objects across time and place (e.g., a bee sting while eating honey during a picnic last weekend). Together, these information stores and their interaction play vital roles in guiding our behaviour and in allowing us to flexibly adapt to variable demands of the environment[2].

Providing extensive support for such functional dissociation within long-term memory stores, the past few decades have witnessed mounting neuroimaging and clinical evidence for separable representations of both semantic and episodic memory in the human brain. Semantic dementia following anterior temporal lobe atrophy, for example, is linked to gradual degradation of conceptual knowledge with relatively spared ability to make new episodic memories[3]. Amnesia consequent on medial temporal lobe lesions on the other hand, has been shown to impair episodic memory, but with largely intact conceptual knowledge[4]. Further expanding this clinical research, neuroimaging evidence from laboratory settings has also revealed extended brain networks for the successful retrieval of information from these dissociable long-term memory stores. Important in the active selection and manipulation of conceptual information, both general semantic[5] and semantic control[6,7] networks have been identified through meta-analytic approaches, revealing regions that span across the inferior frontal cortex, posterior middle temporal gyri, inferior parietal sulcus as well as regions in the cortical midline. In parallel, the retrieval of episodic memory traces has been associated with an extensive posterior medial network that mainly covers the retrosplenial, posterior cingulate cortices, angular gyri and the medial temporal lobe structures[8]. Even in the absence of explicit task demands, parts of these networks form distinct communication routes with associative cortices (e.g., frontoparietal and default mode networks) that potentially make up the foundation of our rich inner mental lives[9].

Despite these long-standing distinctions in the cognitive and neural instantiations of semantic and episodic memory, emerging evidence now calls into question the extent of their separation[10]. Specifically, common cognitive processes are suggested to underlie the large overlap that is observed in the retrieval networks supporting semantic and episodic memory[11]. One core process that is arguably shared across the two memory domains is cognitive control. Generally defined as a goal-directed executive system, cognitive control is postulated to allow the flexible adjustment of prepotent responses to better meet changing and often ambiguous environmental demands. For both semantic[12] and episodic[13] memory, cognitive control is required when dominant memory traces are not sufficiently strong enough to drive appropriate behaviour in an unambiguous manner (e.g., distinguishing between a bee and a wasp for their likelihood to sting). In the case of semantic memory, the automatic retrieval of strongly associated word pairs is consistently linked to activity in brain regions in the posterior parietal cortex[14,15] that partly match the posterior medial network attributed to episodic recollection[16]. Conversely, the left inferior frontal gyrus that is commonly activated in the controlled retrieval of semantic information[17] also shows engagement when participants are asked to retrieve weakly encoded episodic memory traces[18]. Together, this evidence highlights cognitive control as an important aspect of memory retrieval across the two domains with comparable neural instantiations[13]. This, in turn, raises the possibility that the previously reported distinctions in the neural retrieval mechanisms for semantic and episodic memory, which are often based on single studies or isolated meta-analyses conducted across the two sets of literature, might partially reflect quantitative differences in control demands across the laboratory-based tasks that commonly probe these two memory types. In other words, the degree of automatic re-activation versus controlled retrieval processes required to access memory traces under confined experimental settings might constitute an important feature of the long-held distinctions made between the neural retrieval networks of the two long-term memory systems.

Conceptual knowledge about objects and words is acquired over a lifetime. Consequently, common semantic tasks often require the retrieval of well-established knowledge from memory[19]. While there are many features and associations for any concept, only a subset of this information can be probed in particular experimental contexts. For example, conceptual representations can vary in their strength (e.g., bee—honey versus bee—tree), with greater cognitive control required to access weaker associations[15,20]. As semantic tasks typically involve retrieving selective aspects of conceptual knowledge to identify a meaning-based link between items, this requirement to shape retrieval to suit the circumstances, drawing on cognitive control, constitutes a vital aspect of semantic memory tasks. In contrast, episodic memory tasks in laboratory settings, with the exception of autobiographical memory retrieval tasks[21], typically require access to recently acquired memory traces that carry rich contextual content[22]. Difficulties on episodic memory tasks only arise when the cue is inadequate to retrieve the relevant information—for example, when a cue is linked to multiple memory traces, generating interference, or when an episodic memory is weakly encoded as a result of little practice or exposure[13]. However, when retrieval is successful, rich details about internal thoughts and the environment that was present at encoding can be re-instantiated in order to meet task demands. In summary, while it is typical for semantic tasks to manipulate cognitive control demands, episodic tasks employing such manipulations remain limited.

Collectively, these inherent differences in the typical experimental tasks that are used to probe semantic and episodic memory give rise to two alternative hypotheses on the neural mechanisms for long-term memory retrieval: (i) Classical differences in the neural engagement observed across these two memory tasks might reflect distinct retrieval processes for semantic and episodic memory; (ii) alternatively, there might be a common neurocognitive process involved in access to both semantic and episodic memory, but the two isolated sets of experimental tasks across the two domains may place varying demands on more automatic versus controlled forms of retrieval, giving rise to apparent differences in semantic and episodic memory retrieval networks. These alternatives are not easily separable since they require the direct comparison and contrast of the neural responses that underlie controlled retrieval within both domains of long-term memory.

Across two independent experiments, our study used functional magnetic resonance imaging (fMRI) to understand whether there is a common neural system that responds to heightened control demands placed on either memory system. Experiment 1 ($n = 46$) was a multi-session fMRI study in which we acquired measures of neural activity when the same participants were asked to retrieve weak versus strong semantic and episodic memories. We manipulated the strength of conceptual associations between word pairs in the semantic task, and the encoding strength for conceptually unrelated word pairs in the episodic

task. Regions showing a response to both manipulations were candidates for shared processes in the controlled retrieval of the two long-term memory types. Furthermore, Experiment 2 ($n = 140$) was an individual difference study that aimed to examine the association between the intrinsic connectivity of regions that responded to controlled retrieval demands across the two memory tasks in Experiment 1 and performance on an independent set of semantic and episodic memory retrieval tasks administered outside the MRI scanner.

In this work, we identify a shared region in the left inferior frontal gyrus extending towards the anterior insular cortex, which shows greater activity for both semantic and episodic memory retrieval that requires high levels of cognitive control demands. We further compliment this finding by demonstrating that reduced connectivity between the identified region and the ventromedial prefrontal cortex is linked to a selective advantage on tasks that require the retrieval of both weak semantic and episodic memory traces. Together these data are consistent with the hypothesis that common control processes are brought to bear when weak information must be selected from either semantic or episodic memory.

## Results

The main objective of this study was to compare and contrast the neural processes that are engaged in the retrieval of semantic and episodic memory. By introducing a memory strength manipulation (i.e., strong and weak trials) across the two memory types, we were able to assess whether the controlled retrieval of memory hinged upon shared or distinct neural processes. For that purpose, a group of healthy young adult participants ($n = 46$, mean $= 21.31$ years old, SD $= 2.17$, range $= 18–29$, 29/17 female/male) were tested across two separate fMRI sessions that probed semantic and episodic memory retrieval, using 3-alternative forced choice (3-AFC) paradigms. During the semantic 3-AFC task, participants were provided with a word item (e.g., bee) and asked to find the most conceptually associated target word (e.g., sting) amongst three alternative options that included two conceptually unrelated distractors (e.g., plate, atom) (Fig. 1a). In an event-related design, this fMRI task included 40 strong and 40 weak association trials based on the Edinburgh Associative Thesaurus (Supplementary Fig. S1)[23], plus 20 control trials that required a simple motor response.

Subsequently, participants attended a behavioural training session designed to establish strong versus weak episodic associations between two conceptually unrelated words (e.g., apple-flute). Episodic memory strength was manipulated by varying the level of training provided for episodic encoding. On the following day, participants completed an episodic 3-AFC fMRI task with the same parameters employed for the semantic task. The lists of probe-target stimuli across the two fMRI tasks were matched for their psycholinguistic properties (Supplementary Fig. S2). The fMRI data were compared across experimental conditions using general linear models that assessed the main effects of memory strength in the semantic and episodic tasks, as well as conjunctions of memory strength across the two memory types.

As expected, the behavioural results of the participants indicated performance differences in the retrieval of different memory types and for different associative strengths (Fig. 1b). Better performance was observed in strong versus weak association trials across both memory types, as well as for episodic versus semantic decisions (Supplementary Notes S1). In order to account for general task difficulty effects, the average inverse efficiency scores for both semantic and episodic memory 3-AFC tasks were employed as covariates of no interest in the subsequent task fMRI analyses.

**Differential neural engagement in the retrieval of distinct memory types**. Based on prior reports indicating distinctions in the neural instantiations of semantic and episodic memory retrieval[6,8], our initial analysis investigated activation patterns associated with long-term memory type. For that purpose, we compared neural activity differences between semantic and episodic 3-AFC fMRI tasks across all trials, modelling the interaction between memory (semantic versus episodic) and trial types (task versus control).

For semantic versus episodic memory retrieval, the results revealed greater activity centred on the left inferior frontal gyrus, pars opercularis [MNI: −46 12 18, 1466 voxels], extending towards the frontal operculum (Fig. 2a). For episodic versus semantic memory retrieval, however, there was a stronger response in the precuneus extending towards the posterior cingulate cortex [MNI: −6 −72 30, 4345 voxels] as well as bilateral angulari gyri and lateral occipital cortex [MNI: 36 −68 46, 1088 voxels and MNI: −40 −60 40, 534 voxels] (Fig. 2b). Taken together, the results of this initial comparison are in line with prior investigations that indicate a level of separation in the neural basis of semantic and episodic memory retrieval (Fig. 2c, d). For comparison, the results that are uncorrected for general task difficulty effects are provided in Supplementary Fig. S3.

**Common control processes in the retrieval of semantic and episodic memory**. Having observed expected differences in the neural instantiations of semantic and episodic memory, our next objective was to test whether such effects could be related to variations in memory strength and thus the level of control demands required for the retrieval of long-term memory traces. The observation of a shared response in the neural representation of weakly versus strongly associated word-pairs for both memory types would indicate that the neural differences across memory types may be partly rooted in memory strength.

In the semantic 3-AFC task, the retrieval of weakly versus strongly associated word pairs was linked to greater left-lateralised activity centred on the left inferior frontal gyrus (triangularis and opercularis), left inferior temporal gyrus as well as bilateral anterior insular cortex and paracingulate gyrus (supplementary motor cortex) [MNI: −32 22 −2, 13894 voxels]. Concurrent yet less extensive patterns of activity were also observed during episodic memory retrieval. The contrast comparing weakly versus strongly associated word pairs in the episodic 3-AFC task revealed greater activity centred on the left inferior frontal gyrus (triangularis and opercularis) and the anterior insular cortex [MNI: −34 22 2, 877 voxels] (Fig. 3a).

The overlapping patterns of activation during the retrieval of weakly versus strongly associated word pairs were further examined using formal conjunctions across memory types. Controlled retrieval of weakly as opposed to strongly associated word pairs across both memory types elicited a stronger response in the left inferior frontal gyrus and the anterior insular cortex [MNI: −34 21 2, 864 voxels]. When compared to a parcellation of intrinsic brain functional network organisation[24], this conjunction cluster largely overlapped with the frontoparietal control, salience/ventral attention, and default mode networks (Fig. 3b). The reverse contrast of strongly versus weakly associated pairs, which should highlight more unconstrained or automatic retrieval, elicited overlapping activation in the posterior parietal regions, including posterior cingulate/precuneal cortices, middle temporal gyrus, and right angular gyri—sites that almost exclusively fell within the default mode network [MNI: 12 −60 22, 533 voxels] (Supplementary Fig. S4).

Further illustrating the neural response within the identified conjunction region using peristimulus time plots showed sustained activity differences that spanned 1-3 repetition times

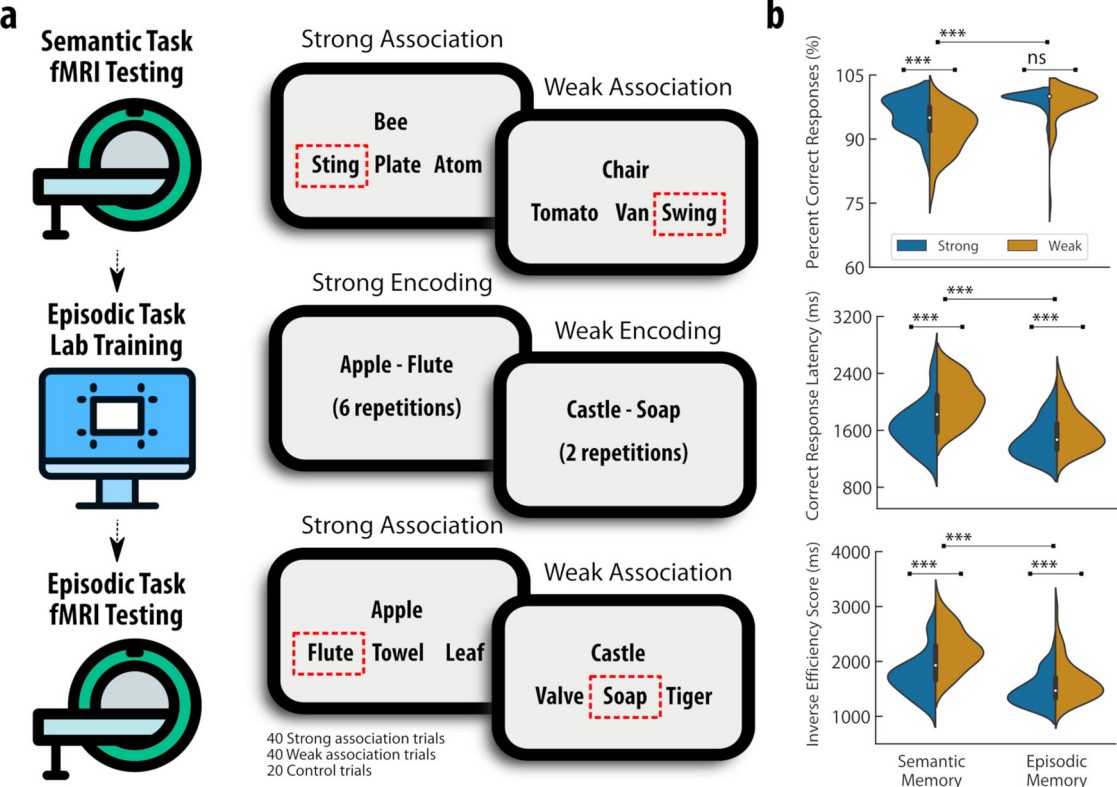

**Fig. 1 Experimental design and behavioural performance during long-term memory retrieval. a** On day one, a group of healthy young adult participants completed a 3-AFC semantic memory retrieval fMRI task in which they were probed with a word (e.g., bee) and asked to select the most conceptually associated target word (e.g., string). On a separate day, participants were trained on pairs of conceptually unrelated words (e.g., apple—flute) using both passive and active encoding. The next day, participants attended an fMRI scanning session in which they were tested during a 3-AFC episodic memory retrieval fMRI task with the same parameters employed for the semantic task. The strength of conceptual associations in the semantic task and level of encoding practice in the episodic task were manipulated (strong versus weak trials). **b** Overall, participants performed better in the retrieval of strongly as compared to weakly associated word pairs for both the semantic and the episodic 3-AFC fMRI tasks. There was also a significant difference in memory type with participants performing better in the episodic in comparison with the semantic memory retrieval task (Supplementary Notes S1). The violin plots illustrate a boxplot with the median (centre white dot), the interquartile range (black bar), the minima/maxima values (thin black line) as well as the kernel density estimation of the underlying distribution. *** denotes $p < 0.001$ in paired t-tests corrected for multiple comparisons. $n = 46$ independent participants examined over two paired fMRI tasks. Source data are provided as a Source Data file.

(TRs) across all participants for both memory types (Fig. 3c, d). Moreover, in addition to the observed main effects at the group level, individual variability in brain activity also depicted significant correlations across the two memory types. Specifically, activity differences in the retrieval of weakly associated word pairs in comparison to control trials was significantly correlated across the semantic and episodic memory retrieval paradigms (partial $r_{\mathrm{p}}$ = 0.27, $p = 0.036$, corrected for age and gender) (Fig. 3e). However, within the same brain cluster no significant correlation was observed in activity differences between the strong versus control trials (partial $r_{\mathrm{p}} = 0.11$, $p = 0.24$, corrected for age and gender). Overall, these results indicate shared neural responses during the retrieval of weakly associated word pairs across the two memory types. As such, our findings suggest that at least part of the memory type differences observed in our initial analysis and in other prior investigations might arise due to underlying differences in memory strength, and thus the level of control demands required to access long-term memory traces.

**Individual differences in the controlled retrieval of long-term memory traces.** Experiment 1 identified a cluster of brain regions centred on the left inferior frontal gyrus and the anterior insular cortex that was commonly engaged in the retrieval of weakly

associated word pairs in both semantic and episodic tasks within the same participant cohort. Converging reports now indicate the predictive power of intrinsic brain networks in explaining trait-like variability in cognitive aptitude[25,26]. Thus, the aim of Experiment 2 was to investigate whether individual differences in the functional interaction of this cluster with any other brain region during resting state fMRI would also be related to performance efficiency in the retrieval of weakly associated word pairs across both semantic and episodic memory domains. In order to answer this question, we employed data from an independent sample of 140 participants. Similar to Experiment 1, participants were asked to retrieve weak semantic associations in a 3-AFC paradigm which involved linking a probe picture to one of three words[27,28] and were also asked to recall unrelated word pairs in an episodic memory task[29].

Taking the left inferior frontal gyrus and anterior insular cortex cluster from Experiment 1 as the seed region-of-interest, we performed a whole-brain connectivity analysis to not only reveal this cluster's intrinsic connectivity architecture, but also to identify areas where the intrinsic connectivity of the seed varied as a function of task performance across the two long-term memory types. First, the results revealed widespread positive connectivity of the seed cluster with regions commonly associated with the frontoparietal, salience/ventral/dorsal attention networks

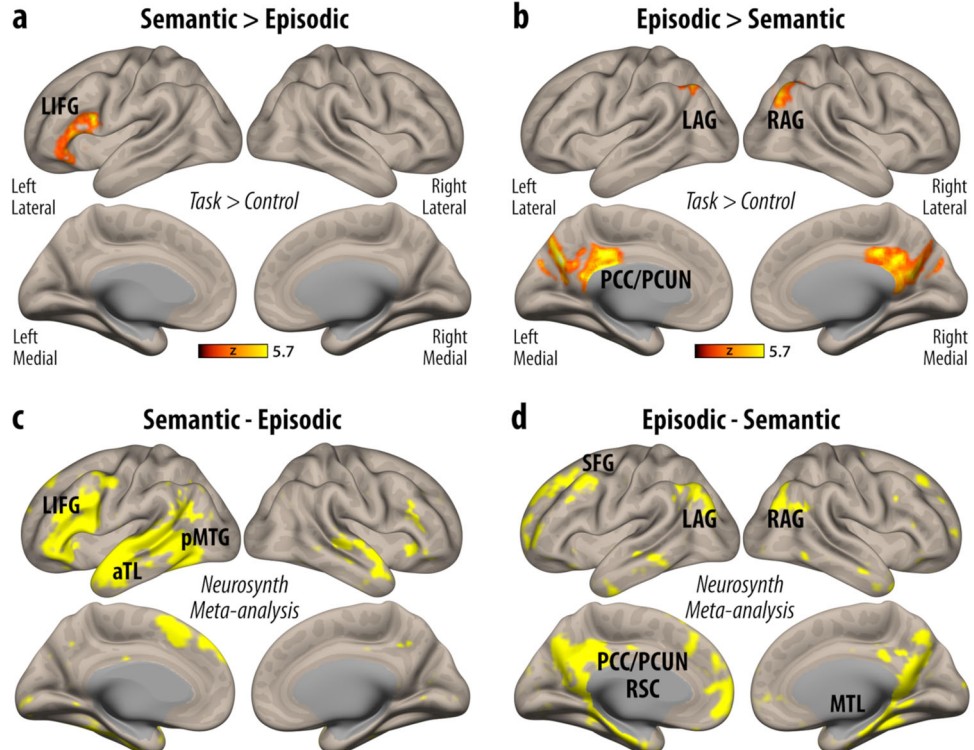

**Fig. 2 Differential brain activity patterns in the retrieval of long-term memory types. a** The comparison of semantic to episodic memory retrieval, across all trial types versus control trials, revealed greater activity in the left inferior frontal gyrus (LIFG). **b** The reverse contrast on the other hand, illustrated greater activity centred on the posterior cingulate/precuneal cortices (PCC/PCUN) and bilateral angular gyri (AG) extending towards the lateral occipital cortex. **c, d** These results spatially overlapped with meta-analytic difference maps obtained from the subtraction of association maps linked to the cognitive terms semantic (n = 1031) versus episodic (n = 488) in the Neurosynth database[57] (aTL = anterior temporal lobe, pMTG = posterior middle temporal gyrus, SFG = superior frontal gyrus, RSC = retrosplenial cortex, MTL = medial temporal lobe). All task-fMRI results were corrected for general task difficulty effects using inverse efficiency scores across memory tasks as covariates of no interest.

as well as visual and auditory networks. Conversely, the same cluster showed negative connectivity (or anti-correlations) with the default mode and motor networks (Supplementary Fig. S5). More importantly, the results revealed that reduced connectivity (or greater anti-correlation) between the left inferior frontal gyrus/anterior insular cortex cluster and bilateral ventromedial prefrontal cortex (vmPFC) [MNI: −8 38 −22, 658 voxels] was associated with better performance (i.e., lower inverse efficiency scores) across both the semantic and episodic memory retrieval tasks (Fig. 4a) (corrected for age, gender and in-scanner head motion). Moreover, individual variation in the connectivity strength of this neural link was neither related to fluid intelligence (partial $r_P = -0.010$, $p = 0.45$), nor selective attention/inhibitory control (partial $r_P = 0.052$, $p = 0.27$) as measured via accuracy-based indices in the Raven's advanced progressive matrices[30] and reaction time to incongruent trials in the Flanker[31] tasks, respectively (Supplementary Notes S2). In addition to this shared neural circuitry, further analysis also revealed a stronger impact of the anti-correlation between the employed seed region and retrosplenial, posterior cingulate cortices on semantic than episodic memory retrieval (Supplementary Fig. S6).

With the aim of further interrogating the cognitive relevance of the identified common neural circuitry the unthresholded statistical map, which related patterns of connectivity to memory performance, was then meta-analytically decoded using the Neurosynth database. The results illustrated that the identified connection between the left inferior frontal gyrus/anterior insula and ventromedial prefrontal cortex was most linked to cognitive terms that spanned semantic, episodic, and autobiographical memory, retrieval, recollection, construction as well as reward,

value, theory of mind, social, and mentalising (Fig. 4b). Finally, with an objective to reveal the intrinsic connectivity pattern of the identified ventromedial prefrontal cortex cluster, this brain region was used as a seed region-of-interest in an additional seed-based functional connectivity analysis. The ventromedial prefrontal cortex cluster's positive connectivity profile revealed large overlaps with the default mode and limbic networks, whereas its negative connectivity profile (i.e., anti-correlation) was most characterised by salience/ventral/dorsal attention and frontoparietal networks (Fig. 4c). Taken together, this body of evidence suggests that intrinsic interactions between left inferior frontal gyrus/anterior insular cortex and ventromedial prefrontal cortex, i.e., functional segregation between hubs of the frontoparietal/salience and default mode networks respectively, relate to individual differences in the controlled retrieval of both weakly associated semantic and episodic memory traces.

## Discussion

The successful retrieval of information from our long-term memory stores constitutes a vital aspect of our ability to ascribe meaning to our environment in order to adaptively guide our behaviour under variable contexts. Although variable control demands are thought to play a key role in the neural instantiation of memory retrieval[2], prior investigations that directly contrasted its impact on different memory types remain scarce. To this end, our Experiment 1 revealed that the controlled retrieval of weakly-associated semantic and weakly-encoded episodic memory traces elicited shared engagement of a brain cluster located on the left inferior frontal gyrus and anterior insular cortex. Experiment 2

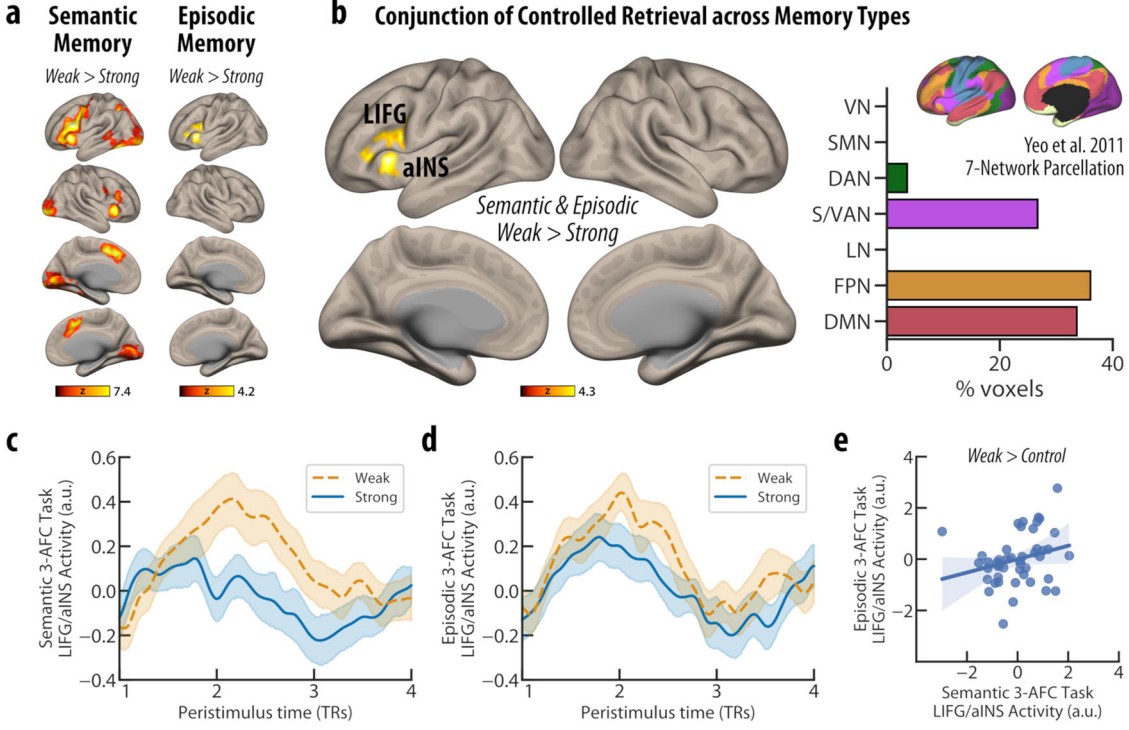

**Fig. 3 Shared brain activity patterns associated with controlled retrieval across long-term memory types. a** For both semantic and episodic 3-AFC fMRI tasks, shared neural responses were observed in the retrieval of weakly versus strongly associated word-pairs. **b** A formal conjunction analysis illustrated that for the weak > strong contrast, left inferior frontal gyrus (LIFG) and anterior insular cortex (aINS) showed greater activity for both memory types, which largely overlapped with the frontoparietal, salience/ventral attention, and default mode networks (based on the Yeo 7-Network parcellation[24]) (VN = visual network, SMN = somatomotor network, DAN = dorsal attention network, S/VAN = salience/ventral attention network, LN = limbic network, FPN = frontoparietal network, DMN = default mode network). **c, d** The peristimulus time plots (for illustrative purposes only) showed sustained activity differences between weak versus strong trials (in comparison to implicit baseline) that spanned 1-3 repetition times (TRs) across all participants for both memory types. While curved lines represent the best fit to a smoothing spline over mean activity values, shaded areas reflect 95% confidence intervals. **e** LIFG/aINS activity differences during weak versus control trials between semantic and episodic memory tasks were significantly correlated across participants (partial $r_p$ = .29, $p$ = 0.036, correcting for age and gender). Straight-line represents the best fit, while shaded areas illustrate 95% confidence intervals. No such correlation was observed for activity differences between strong versus control trials for this cluster of brain regions across the two memory types (partial $r_p$ = .11, $p$ = .24, correcting for age and gender). Source data are provided as a Source Data file.

then demonstrated that individual variation in the functional interactions of this cluster at rest with the ventromedial prefrontal cortex was associated with differences in the performance efficiency of controlled retrieval for both semantic and episodic memory. Overall, this converging body of results robustly indicate that memory strength plays a pivotal role in the neural instantiations of long-term memory types and that previously reported semantic-episodic dissociations might at least be partially rooted in the level of cognitive control demands required for memory retrieval.

Two distinct streams of literature have long underlined differential brain networks associated with the retrieval of semantic and episodic memory. While the semantic control[6] and general semantic[5,7] networks are suggested to constitute core components of semantic cognition[12], the posterior memory network[8] is argued to support episodic memory retrieval. As such, semantic and episodic memory have been largely studied in isolation as separate systems, with contrasting representational structures and associated retrieval mechanisms. However, an alternative view of a unitary memory system is re-gaining attention within cognitive neuroscience, in which semantic and episodic categorisation is postulated to represent two ends of a continuum of memory expression along dimensional attributes (e.g., abstraction, familiarity, salience)[10,11,32]. Earlier studies testing this hypothesis found common neural responses across the cerebral cortex and the medial temporal lobe structures in the retrieval of

autobiographical, episodic, and semantic memory as well as distinctions that were interpreted as reflecting specific properties of the retrieved memory traces[33–35]. In parallel, more recent perspectives argue for large overlaps in the neural instantiation of long-term memory traces with propositions made for the potential existence of a "core recollection network"[5]. Further experiments highlight the interdependence of these two memory systems in successful memory retrieval[36,37] with recent clinical neuropsychology investigations calling into question the historical attribution of different memory types to disease-specific brain regions[38,39]. Collectively, this resurging view advocates for blurred lines in the neural instantiations of semantic and episodic memory retrieval with different modes of functioning within a single memory system suggested to give rise to the long-held categorical differences in memory types.

To this end, the complementary results of our two experiments suggest that one continuum along which the neural instantiations of memory retrieval may rely upon is quantitative differences in the strength of memory probed in experimental tasks that assess different domains of memory. In other words, the level of automatic versus controlled retrieval required to access memory traces under variable contextual goals may give rise to the observed differences in the neural instantiations of these two memory types. Converging evidence now highlights the vital role that the prefrontal cortex plays in the cognitive control of memory, especially in the selection of relevant and interference-suppression of irrelevant memory traces[40]. In the

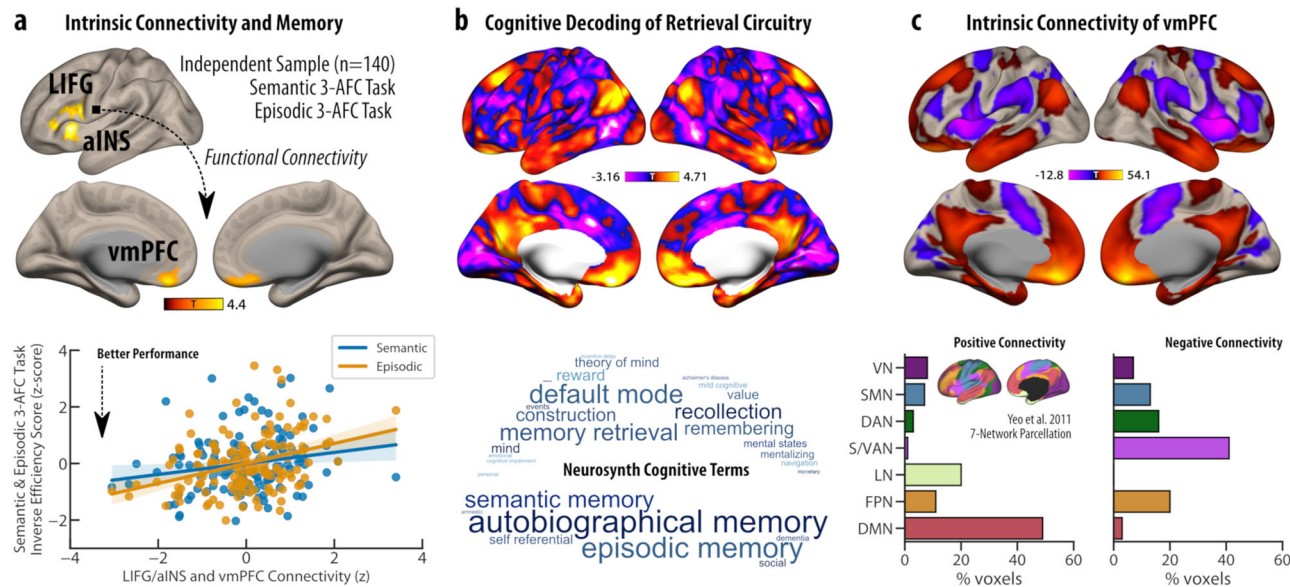

**Fig. 4 Common neural circuitry linked to the controlled retrieval of semantic and episodic memory. a** A large cohort of 140 participants underwent a nine-minute long resting-state fMRI scanning. In the analysis of this Experiment 2 dataset, the significant cluster that was linked to the retrieval of weak versus strong associations across both memory types in Experiment 1, was used as the seed region-of-interest in a functional connectivity analysis. Across participants, lower positive connectivity (greater anti-correlation) of the left inferior frontal gyrus/anterior insular cortex (LIFG/aINS) cluster to the bilateral ventromedial prefrontal cortex (vmPFC) was related to better performance in the retrieval of both weakly associated semantic and episodic word pairs, measured outside the scanner (corrected for age, gender and head-motion). The scatter plot is shown for illustrative purposes only. While the straight lines represent the best linear fit over individual values, shaded areas illustrate 95% confidence intervals. Individual variation in this neural circuitry was not related to fluid intelligence (partial $r_p = -0.010$, $p = 0.45$), or selective attention / inhibitory control (partial $r_p = 0.052$, $p = 0.27$). $n = 140$ independent participants examined over one experiment. **b** The unthresholded results from this regression model were meta-analytically decoded for cognitive terms using the Neurosynth database. The top 100 terms are presented in a word cloud in which text size corresponds to the strength of association (anatomical terms were removed). **c** In a separate seed-based functional connectivity analysis, the positive connectivity ($r > 0$) of the vmPFC revealed large overlaps with the default mode and limbic networks, whereas negative connectivity ($r < 0$) was characterized by salience/ventral/dorsal attention and frontoparietal networks, based on the Yeo 7-Network parcellation scheme (VN = visual network, SMN = somatomotor network, DAN = dorsal attention network, S/VAN = salience/ventral attention network, LN = limbic network, FPN = frontoparietal network, DMN = default mode network). Source data are provided as a Source Data file.

domain of semantic memory, control manipulations have been consistently shown to elicit activation within a distributed network including the left inferior frontal gyrus, which is partially overlapping with yet partially distinct from the 'multiple-demand' regions that respond to control demands across tasks[6,17,27]. In comparison to such task-agnostic control demands, the left-lateralised findings across the two memory domains in our study point towards selective cognitive control processes that may be ascribed to the retrieval of memory. In line with this perspective, in a transcranial magnetic stimulation study, the disruption of the left inferior frontal gyrus has been shown to selectively impair performance on the retrieval of weakly associated word pairs, with no such influence observed over the retrieval of strongly associated word pairs or strength manipulation in a perceptual task[17]. Likewise, a meta-analysis over a wide variety of semantic tasks reported greater left inferior frontal gyrus engagement in comparison to equally difficult phonological tasks[6]. Furthermore, similar left-lateralised results have also been observed across studies contrasting weakly-encoded versus strongly-encoded episodic memory traces[18] and the recollection of memory sources from experience[41]. Collectively, this evidence underlines the specific contribution of this left-hemispheric region to the successful retrieval of both semantic and episodic memory over-and-above those observed due to general task difficulty effects.

Providing further evidence for the importance of this control region in the retrieval of both memory types, the results of our Experiment 2 indicated that its functional interactions with the ventromedial prefrontal cortex explained individual variation across the retrieval of both weakly associated semantic and weakly

encoded episodic memory traces. Specifically, this trait-like variation across an independent cohort of 140 participants revealed that reduced positive connectivity (or greater anti-correlation) of the left inferior frontal gyrus and anterior insular cortex cluster to the bilateral ventromedial prefrontal cortex was associated with better performance in the retrieval of weakly associated picture-word pairs, and the immediate recall of weakly encoded episodic word pairs. While the meta-analytic decoding of this circuitry revealed strong links with terms that encompassed semantic, episodic, autobiographical memory, memory retrieval, recollection, construction, further seed-based analysis confirmed that the identified ventromedial prefrontal cortex region is a core area within the default mode network. This finding is consistent with the theoretical insight that while there may be complementary representational systems—with anterior temporal lobe representing commonalities across experiences and hippocampus supporting pattern completion—retrieval pathways for semantic and episodic memory may be partly overlapping, such that stronger and more coherent experiences across both types of memory may be supported by regions within the default mode network, while weaker and less coherent experiences recruit control regions[6,11,42]. Hence, greater separation of the intrinsic connectivity linked to these component processes may reflect greater control of memory over automatic aspects of cognition[27,42]. To this end, emerging evidence now highlights the important role played by the vmPFC in schema-mediated memory retrieval[43], in contrast to the common attribution of the left inferior frontal gyrus to controlled retrieval processes[13]. Nevertheless, further task-based studies with causal methods (e.g., transcranial

magnetic stimulation) and finer time resolution (e.g., magnetoencephalography) will be required to understand the directional exchange of information in the service of controlled memory retrieval.

Notable limitations should be considered when interpreting the results of our study. First, although the retrieval paradigms in our Experiment 1 were designed in line with the current literature and in a manner that would categorically separate semantic and episodic memory, perfect isolation of memory retrieval across the two domains could not be guaranteed. Conceptual information could have been retrieved during the episodic memory task. Conversely, rich autobiographical and episodic memory could have been re-activated during the performance of the semantic memory task. Hence, a common strength manipulation across both paradigms was necessary to deduce the effects of control demands that are largely independent of domain categorization. Nevertheless, further research with variable paradigms and stimulus modalities will be required to assess the reliability of our results to operational categorization of different memory types. Second, behavioural results indicate task difficulty differences across the two experimental paradigms. Although this nuisance factor was taken into account during data analysis, further studies with greater balance in task difficulty across the two memory domains may identify even more extensive overlap in the neural mechanisms of controlled long-term memory retrieval. Finally, a data-driven analysis approach was employed in Experiment 2 in which we selected a region of interest based on the shared activity patterns observed in our Experiment 1. Despite providing complementary findings across the two experiments, this approach was limited in identifying regions beyond the left inferior frontal gyrus and anterior insular cortex that may also relate to behavioural performance. Future studies that employ whole-brain methods such as connectomic fingerprinting[26] may reveal wider functional interactions and topological configurations that predict individual differences in long-term memory retrieval.

Notwithstanding these limitations, the two independent experiments employed in this study provide complementary evidence for common neural processes that support controlled retrieval of both semantic and episodic memory. Consequently, the classical neural distinctions that are reported in the literature between semantic and episodic memory tasks may represent two extreme ends of a continuum of long-term memory expression, in which memory strength plays a pivotal role, as opposed to qualitatively distinct semantic and episodic memory retrieval systems. In line with emerging theoretical and empirical studies[38,39,44,45], this alternative way of conceptualising memory in the human brain may potentially have important consequences for both theory and neuro-rehabilitation of memory disorders.

## Methods

**Participant demographics.** For both Experiment 1 and 2, ethical approval was obtained from the Department of Psychology and York Neuroimaging Centre, University of York ethics committees. All participants were briefed about the aims and objectives of the experiments before providing informed consent to take part in these studies. For Experiment 1, a total of 47 undergraduate or postgraduate students were recruited who were all right-handed, native English speakers with normal to corrected-to-normal vision. Under the same inclusion criteria, 169 students were recruited for Experiment 2. Participants received monetary reward of £10 per hour for their participation in these studies.

As per the exclusion criteria, none of the participants had any prior history of psychiatric or neurological disorders, incompatibility for MRI scanning, severe claustrophobia and anticipated pregnancy or drug use that could alter cognitive functioning. In Experiment 1, one participant had incomplete data which was excluded from further analysis. In Experiment 2, a total of 29 participants were excluded due to missing data and excessive motion inside the scanner based on the extensive head-motion correction procedures described below. Consequently, the final group for Experiment 1 consisted of 46 participants (mean = 21.31 years old, SD = 2.17, range = 18–29, 29/17 female to male ratio), whereas the final number of

participants included in Experiment 2 was 140 (mean = 20.70 years old, SD = 2.37, range = 18–31, 83/57 female to male ratio), who fully completed all the required neuroimaging-based and behavioural assessments.

### Experiment 1

*Experimental design and paradigms.* For Experiment 1, the study design required a total of three visits on three separate days: two visits to the York Neuroimaging Centre for fMRI scanning during the performance of semantic and episodic memory tasks, and one visit to the behavioural laboratory at the Department of Psychology, University of York for episodic memory training.

Day One: Participants attended a neuroimaging session in which they performed a 3-AFC semantic memory task that aimed to assess the retrieval of semantically associated conceptual information. Modelled on our prior investigations[27], participants were shown a probe (e.g., bee) and three response options in words, one of which constituted a target with a conceptual link to the probe (e.g., sting), while the other two were distractors with no such conceptual link (e.g., plate, atom). The strength of semantic association between the probe and target was manipulated, operationally categorized into strong versus weak semantic associations based on the Edinburgh Associative Thesaurus[23]. The distribution of conceptual strengths for the two categories are provided in Supplementary Fig. S1.

Day Two: Participants attended an episodic memory training session at the behavioural laboratory. The aim of this session was to train participants on encoding pairs of conceptually unrelated words. For that purpose, first, a passive encoding block was employed in which each word-pair was displayed in the middle of the screen (words were separated with a dash) for five seconds, interleaved by one-second fixation cross. Immediately following, an active encoding session was administered in which the participants were given the probe word and asked to correctly type out the target word within 10 s. A corrective feedback was provided indicating whether the response was correct or wrong for three seconds. The strength of association was manipulated based on the level of training provided. For strongly encoded word-pairs, the active encoding session was repeated three times, whereas, for weakly encoded-word pairs, the session was given once. The overall battery was repeated twice for both trial types.

Day Three: Participants attended a neuroimaging session in which they performed a 3-AFC episodic memory task that aimed to assess the retrieval of strongly and weakly encoded episodic associations. Specifically, participants were shown the probe word and three response options in words, one of which constituted the target that the probe was paired with during episodic training, while the other two were unlinked distractors. The strength of encoding was manipulated by the level of training provided on the previous day i.e., a total of six repetitions of a combination of active and passive encoding sessions for strong, and a total of two repetitions for weak episodic associations.

For both the semantic and episodic 3-AFC tasks, the probe was displayed in the middle of the screen at the same time as the three options, which were displayed below. The position of the target and two distractors were randomized. Participants were given four seconds to respond, after which a fixation cross was displayed, jittered in duration between 1.5 and 3.5 s in 500 ms intervals. A total of 40 strong and 40 weak trials were employed with 20 control trials in which the participants were simply shown XXX on all probe and response option positions and asked to press any one of three buttons. The complete list of words was divided into two groups which were randomised across subjects and tasks. All probe-target pairings were matched for psycholinguistic properties (Supplementary Fig. S2) including concreteness, familiarity, imageability and number of letters based on the MRC psycholinguistic database[46], and lexical frequency extracted from the SUBLEX-UK database[47]. The total duration for both tasks was 10.83 min. The stimuli across all behavioural and neuroimaging tasks employed in this study were visually presented using PsychoPy2 (Version 1.82)[48].

*MRI data acquisition.* Neuroimaging sessions were carried out at the York Neuroimaging Centre, York with a 3 T GE HDx Excite MRI scanner using an eight-channel phased array head coil. The imaging parameters for the high-resolution structural scan with 3D fast spoiled gradient echo were as follows: TR = 7.8 s, TE = 3 ms, flip angle = 20°, matrix size = 256 × 256, 176 slices, voxel size = 1.13 × 1.13 × 1 mm. The functional MRI data for both semantic and episodic 3-AFC tasks was collected using single-shot 2D gradient-echo-planar imaging sequence (TR = 3.0 s, TE = 18.9 ms, flip angle = 90°, matrix size = 64 × 64, 60 slices, voxel size = 3 × 3 × 3 mm³, 217 volumes). With the aim of improving co-registration, a fluid-attenuated inversion recovery (FLAIR) sequence with the same orientation as the functional scans were also acquired.

*MRI data analysis.* The preprocessing of the task fMRI data involved skull-stripping, motion correction via MCFLIRT, slice-timing correction with Fourier space time-series phase-shifting, high-pass filtering (sigma = 100 s), normalisation to the MNI-152 template space using FSL FLIRT (Version 6.0) as well as skull-stripped FLAIR and high resolution T1-weighted structural images, and spatial smoothing with a 5 mm full-width-half-maximum (FWHM) Gaussian kernel.

The preprocessed task fMRI data was then modelled using the general linear model (GLM) with the onsets and durations (4 s) of all trial types (i.e., strong, weak, control – correct trials only). Higher-level statistical contrasts of strong > weak,

weak > strong and task (all trials) > control was assessed using a mixed-effects approach with FLAME. Conjunction analyses across semantic and episodic tasks were carried out with the easythresh_conj command from FSL that relies on the minimum statistic of the conjunction null[49]. For all GLMs, standard motion parameters (3 rotations and translations), their temporal derivatives and squared versions were added as potential motion confounds. Furthermore, motion outliers were identified using the DVARS method and included in the model as a covariate of no-interest[50]. No significant difference in head-motion was observed between the two task conditions (Supplementary Fig. S7). In addition, the mean inverse efficiency scores for each participant (i.e., reaction time divided by the percentage of incorrect responses) across both semantic and episodic memory tasks were added as a covariate of no-interest in order to correct for general tasks difficulty effects. All results were cluster-corrected for multiple comparisons using the family-wise error (FWE) detection technique at the 0.05 level of significance (cluster-forming threshold z > 2.6). For illustration purposes, peristimulus time plots were generated based on activity differences in the contrast of interest, which were averaged and interpolated across events for each participant using fitted smoothing spline curves.

### Experiment 2

*Experimental design and paradigms.* In Experiment 2, we employed a large-scale dataset that was previously collected and utilised in our prior investigations[51]. In this experiment, participants first attended a testing session outside the scanner at the behavioural laboratory, in which they were asked to perform a battery of cognitive tasks and psychometric assessments that examined various aspects of cognition, including semantic and episodic memory retrieval. Based on the current study objectives on the assessment of shared neural responses across controlled retrieval of weakly associated memory traces, we selected two conditions from the battery that most closely matched the fMRI tasks employed in Experiment 1. In addition, we employed indices from validated tests of fluid intelligence (Raven's advanced progressive matrices)[30] and selective attention/inhibitory control (Flanker task)[31] as control measures (Supplementary Notes S2).

For semantic memory, a 3-AFC task was utilised, in which participants were asked to select from three response options in words (e.g., cat, car, shampoo) one of which was a weakly associated concept that most closely matched a picture probe (e.g., dalmatian)[27,28]. For episodic memory, behavioural responses from a paired-associates task, similar to the one employed in episodic memory encoding in Experiment 1, were examined. Participants were first shown a list of 40 word-pairs with no semantic associations (e.g., castle-soap) for 5 s, interleaved by a one-second-long fixation cross. Following this passive encoding period, participants were asked to actively recall the second word after the presentation of the first word-pair by typing out their answers within a 12 s window for each trial[29]. Corrective feedback was provided in which participants had to reach 60% correct responses within three repetitions of the full wordlist, which matched the manipulation for weak-encoding provided in the episodic 3-AFC task in Experiment 1. Immediately following the encoding session, an episodic recall phase was administered without feedback in order to assess weakly encoded episodic memory retrieval.

*MRI data acquisition.* Following the same imaging protocol from Experiment 1, participants were first scanned with a 3D fast spoiled echo sequence to obtain structural images. In addition, a nine-minute resting-state fMRI scan was performed with a single-shot 2D gradient-echo-planar imaging sequence (180 volumes). For the duration of the resting state scanning, participants were asked to keep their eyes open and to focus on a fixation cross presented in the middle of the screen.

*MRI data analysis.* Following the removal of the first three functional volumes to achieve steady-state magnetisation, the remaining resting state fMRI data was slice-time and motion corrected, co-registered to the high-resolution T1 image and normalised to the Montreal Neurological Institute (MNI-152) space utilising the unified segmentation–normalization framework[52]. Finally, an 8 mm full width at half maximum (FWHM) Gaussian kernel was used for spatial smoothing.

An extensive set of motion-correction and denoising procedures were employed, comparable to those reported in the literature[53]. In addition to the removal of six realignment parameters and their second-order derivatives using GLM, a linear detrending term was applied as well as the CompCor method that removed five principal components of the signal from white matter (WM) and cerebrospinal fluid (CSF)[54]. Moreover, the functional volumes influenced by excessive head motion were identified and scrubbed based on the conservative settings of motion greater than 0.5 mm and global signal change larger than z = 3. Participants who had more than 15% of their data affected by motion were excluded from further analysis. In addition, the composite motion score (i.e., percentage of invalid scans) for each participant was also added as a covariate in group-level analyses to further account for the potential influence of head motion on functional connectivity estimates. No significant correlation was observed between in-scanner head motion and covariates of interest utilised in subsequent analyses (Supplementary Fig. S7). Global signal regression is known to introduce spurious anti-correlations with recent reports indicating its predictive power for explaining cognitive factors, and thus it was not utilized in our analysis[55]. Finally, a

band-pass filter between 0.009 Hz and 0.08 Hz was employed in order to focus on low frequency fluctuations.

Following this procedure, we performed a data-driven seed-based functional connectivity analysis using the significant cluster from the conjunction of weak > strong contrasts across semantic and episodic memory tasks in Experiment 1 as a seed region-of-interest. For each participant, average BOLD time series obtained from the binarized version of the significant control cluster from Experiment 1 was correlated with time courses from the rest of the brain in order to obtain individual and group-level connectivity maps. Linear regression with seed-based connectivity was performed in which inverse efficiency scores across both semantic and episodic memory tasks were included as the variables of interest. Age, gender and composite motion score (i.e., percentage of invalid scans identified during the scrubbing procedure) were added as nuisance variables in the model. All reported clusters were corrected for multiple comparisons using FWE at the 0.05 level of significance (uncorrected at the voxel-level, 0.005 level of significance). Significant results are displayed on a scatter plot for visualisation purposes, using the Seaborn Python package[56]. Finally, the unthresholded connectivity maps from the above analysis was meta-analytically decoded using the Neurosynth database (https://neurosynth.org/decode/)[57]. The top 100 cognitive terms were displayed in a word cloud, from which anatomical terms were removed.

**Reporting summary**. Further information on research design is available in the Nature Research Reporting Summary linked to this article.

## Data availability

The datasets generated and/or analysed during the current study are not publicly available due to institutional regulations, ethics, and confidentiality agreements, but are available from the corresponding author on reasonable request. Unthresholded statistical maps (z-maps) from the task-based and resting-state fMRI portions of this study are publicly available at https://identifiers.org/neurovault.collection:8431. Source data are provided with this paper.

## Code availability

The preprocessing, denoising, and statistical analyses of all task-based and resting state fMRI data were performed based on routines from the FMRIB Software Library (FSL) (Version 5.0.11) (https://fsl.fmrib.ox.ac.uk/fsl/fslwiki), SPM (Version 12.0) (http://www.fil.ion.ucl.ac.uk/spm/), Conn functional connectivity toolbox (Version 17.f) (https://www.nitrc.org/projects/conn) and MATLAB platform (Version 16.a) (https://uk.mathworks.com/products/matlab.html). Python code to regenerate figures from the Source Data file using Seaborn routines are available from the authors on request.

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

## Acknowledgements
We thank all staff at the York Neuroimaging Centre for their assistance with MRI scanning. This study was supported by the European Research Council (Project ID: 646927). Additionally, DV was funded by the National Natural Science Foundation of China (No. 31950410541), the Shanghai Municipal Science and Technology Major Project (No. 2018SHZDZX01), and ZJLab. The icons in Fig. 1 are based on resources from Flaticon.com.

## Author contributions
D.V., J.S., and E.J. equally contributed to the conception and design of the work. D.V. collected and analysed the data, and wrote the first draft of the manuscript, which was then revised by D.V., J.S., and E.J.

## Competing interests
The authors declare no competing interests.
