## [Peer Review File · Nature Communications]

Reviewer #1 (Remarks to the Author):

This is an interesting study on a timely and important topic, namely declarative memory and similarities between semantic and episodic memory in particular. The paper was clear, well written and used an elegant design to compare semantic and episodic retrieval across different levels of association strengths (and thus retrieval demands).

The finding of similar control networks is somehow expected based on the authors previous publications. More generally, it is uncontroversial to classify the control network as related to executive functions (and executive control over retrieval) and thus not necessarily distinct for episodic and semantic memory, as per the authors' current findings and consistent with recent literature. The authors could cite recent works on this topic coming to similar conclusions on the similarity in the neural bases of semantic and episodic memory, including executive control over retrieval (e.g., Keane, Bousquet, Wank, Verfaellie, 2019; Renoult, Irish, Moscovitch, Rugg, 2019; Duff, Covington, Hilverman, Cohen, 2020; Solomon & Schapiro, 2020)

I was not totally clear with the idea presented in the introduction that typical semantic and episodic tasks are not matched in control processes (lines 87-89). The authors suppose that semantic tasks would typically require a high degree of control, but not episodic memory tasks (Line 106-108): "However, when retrieval is successful, rich details about internal thoughts and the environment that were present at encoding can be automatically re-instantiated in order to meet task demands." If words automatically activate their meanings (e.g., Neely & Kahan, 2001) then in many typical semantic tasks (e.g., sentence verification, concrete-abstract classification, living/non-living categorization, etc) little cognitive control may need to be applied; a similar point made by the authors for episodic memory.

Line 297-299 (see also introduction lines 80-82). I did not really understand why the authors compared the semantic control network to the posterior memory network. Would it not make more sense to compare the semantic retrieval network to the episodic retrieval network (that some may call posterior memory network) that recent literature mentioned above show had clear overlap?

One of the main conclusions of the paper is that differences in neural correlates usually reported between semantic and episodic memory would be due to differences in cognitive control (e.g., lines 140-142: "Thus, classical distinctions in the semantic and episodic retrieval networks in the literature may reflect differences in the cognitive control requirements of commonly used long-term memory tasks"; see also discussion lines 291-294). It was not clear to me how this would work. As mentioned above, quite a few investigators came to the conclusion recently that semantic and episodic retrieval networks are very similar. Typical differences in neural correlates between semantic and episodic memory may not be easily interpreted in term of cognitive control. For instance, quite a few investigators would think (consistent with literature presented in this manuscript) that the hippocampus has a clear association with episodic memory (and perhaps less so for semantic memory) and that the anterior temporal lobe has a clear association with semantic memory (and perhaps less so with episodic memory): how do these differences can be accounted in term of the control network?

The point made line 93-103 that semantic tasks typically investigate knowledge learned over many years, while lab-based episodic memory tasks investigate recently learned associations is a fair one, but is not valid for the rich literature of autobiographical memory studies, where one often probes semantic and episodic details for the same memories. The core recollection network observed by these tasks is very similar (or identical) to that observed in lab-based studies (e.g., Moscovitch et al., 2016).

Reviewer #2 (Remarks to the Author):

Vatansver and colleagues conducted two well-powered fMRI studies investigating the possibility

that both episodic and semantic retrieval tasks rely on a common cognitive control process. In the first study, subjects (N=46) were each scanned twice as they performed two carefully matched tasks of episodic and semantic retrieval. As with many prior studies, the authors found some regions that were more active for semantic retrieval and others more active during episodic retrieval. But the core analyses focused on contrasts between trials with weak vs. strong memories (either based on pre-experimental associative strength [semantic strength] or number of study trials [episodic strength]). The key finding is that one brain region – a large swath of left inferior frontal gyrus and anterior insula – showed increased activity for weak memories during BOTH episodic and semantic retrieval. The authors interpret this overlap as indicating that the region makes a common contribution to mnemonic control processes, and that its function should not be attributed to a particular memory system. The authors then use this region as a seed for a whole-brain resting state functional connectivity analysis conducted on an independent fMRI dataset (N=140). Their main finding is that the strength of this region's intrinsic anti-correlation with the ventromedial prefrontal cortex was correlated with individual differences in performance on both the semantic and episodic memory tasks.

Understanding the similarities and differences between episodic and semantic memory is a hot topic in the field right now, and this paper's intriguing set of results will be read with interest by many memory researchers. I was impressed with the effort that went into collecting such large fMRI datasets (including one experiment that involved sessions across three separate days), and I thought the perceptually matched episodic and semantic retrieval were cleverly designed. Generally speaking, the fMRI data analyses were thoughtfully executed and the well-crafted figures provide informative visualizations of the findings. That said, I have some concerns about the interpretability of the findings, and there are issues with a few of the analyses that I think should be addressed.

1. The core finding of the paper seems to be that the left inferior frontal gyrus (LIFG) is involved in both episodic and semantic retrieval, with the implication that its role has more to do with cognitive control processes that facilitate access to weakly associated memories than with the recovery of actual episodic or semantic content per se. However, an alternative interpretation of the results is that this region's contributions are heavily semantic in nature, and that many episodic memory tasks (especially those like the present study that involve building memorable associations between seemingly unrelated words) have a strong conceptual processing component. Indeed, a recent review by Renoult, Irish, Moscovitch, Rugg (TiCS, 2020) states that: "consistent with the notion that the left inferior prefrontal cortex supports semantic control processes... it has been reported that the region is more active during episodic memory tests that place high, as opposed to low, demands on the processing of retrieval cues."

In the context of the present experiment, the strong episodic trials assess memory for unrelated word pairs that had been studied six times. If a subject had six opportunities to learn the pairing of APPLE-FLUTE, they probably will have built up a richly elaborated semantic connection between these two concepts, such as visualizing someone play a flute under an apple tree). Word pairs that were studied only twice (weak episodic trials) probably also involved the generation of semantic linkages between the referents, but these associations may have been more tenuously constructed and thus will place greater demands on semantic processing during retrieval. The episodic task was trivially easy for subjects, yielding 99% correct on strong trials and 98% correct on weak trials, so subjects were almost always able to successfully recall which of the three probe words had been paired with the target word. The extra time that participants took to respond on weak trials may have been used to mentally simulate a semantic connection between the weakly linked words and evaluate whether that connection felt like one they had thought of the prior day. My general point here is that presence of overlapping LIFG activation in the weak > strong contrasts for the episodic and semantic tasks does not rule out the possibility that differential semantic processing is responsible for both contrasts. Perhaps the authors could partially rule this out by running a task-related functional connectivity analysis and showing a differential profile of connectivity of the LIFG depending on whether episodic vs. semantic knowledge needed to be accessed. But even such a demonstration would not adjudicate between the authors' proposed strength-based account and an account that posits that the overlap is due to common demands for semantic processing on the weakly associated trials of both tasks.

2. In Fig 3E, the authors show a scatterplot of the correlation between participants' LIFG activation level during weak trials of the episodic and semantic tasks. The take-home point of this analysis is not entirely clear, as little is said about why it was done or what it means. But the analysis itself seems potentially problematic. Rather than computing weak > strong activation difference scores for each subject and then correlating those across tasks, the authors are correlating the single condition parameter estimates for weak trials (vs. the implicit fixation baseline). Thus subjects with more robust task-related BOLD responses (perhaps attributable to better neurovascular coupling, more alertness, more caffeine, or less head movement, etc.) will tend to show higher activation parameter estimates in both tasks. To rule this out and show that the effect is specific to trials with weakly associated stimuli, the authors should demonstrate that no such correlation exists between the episodic and semantic "control task trials", or between strong episodic and strong semantic trials.

3. The authors show hemodynamic timecourse plots in Fig 3C and 3D to illustrate "a clear separation in brain activity associated with the weak versus strong association trials for both memory types", but this is circular claim, since the region-of-interest was defined precisely on the virtue of it showing this weak > strong effect. If the timecourse plots have some additional informational value that the authors believe is worth conveying to readers, they should have a clearly stated circularity warning (e.g., "shown for illustrative purposes only").

4. I have a similar concern about inferential circularity that pertains to the scatterplot shown in Fig 4A. The authors conducted a whole-brain seed-based connectivity analysis and found a significant connectivity-behavior relationship in the vmPFC. They then show us a scatterplot depiction of this correlation from that very region, which was defined entirely on the basis of it showing this effect. Unless I am missing something about the way that this scatterplot figure was derived, it is a misleading figure that provides little informational value and should be removed.

5. More broadly, I struggled to understand what was learned about the nature of episodic and semantic memory by doing this analysis of individual differences in resting state connectivity. The authors show us that the intrinsic anti-correlation between the LIFG/aINS and the vmPFC is stronger in those individuals with better memory performance. But how do we even know that this effect has anything to do with memory ability per se? Because there is no non-memory control task, it is possible that individuals who are better performers in general (perhaps because of higher motivation, attentional focus, decision-making, intelligence, and/or reading ability) would also be the ones with the highest anti-correlation between these two regions. Even if the effect was proven to be specific to memory ability, the authors should say more about what they think this effect means (especially in light of the fact that a big portion of their LIFG/aINS cluster is designated as a DMN region).

6. The authors feature the vmPFC as being the only region that showed a significant relationship between its connectivity with the LIFG/aINS and behavioral performance on the two tasks. However, Fig 4B shows an unthresholded view of the same map, and there seem to be comparably robust connectivity-behavior relationships in number of other default mode network regions, including bilateral angular gyrus, posterior cingulate cortex, and superior frontal gyrus. Did none of those other effects achieve significance?

6. The use of Neurosynth to label the relative network membership composition of activation clusters or whole brain maps is a cool addition to the paper, but I would like the authors to elaborate more on what is learned by knowing these relative network membership compositions for each analysis. For instance, the major focus of this paper is a large LIFG/aINS cluster that seems to span across three different functional networks. Do the authors suspect that different components of this large cluster contribute in different ways, depending on their network membership?

7. Although I found it visually appealing, it is ultimately misleading to print the Neurosynth word cloud terms on top a brain in Fig 4B, since it gives a false impression of region-to-function mapping.

Minor comments:

Was the fMRI data analysis limited to correct trials only?

What was the cluster extent threshold requirement for the whole brain analyses?

How did the authors get such fine grained temporal resolution in their timecourse plots when the TR was 3 sec?

p. 1, In the abstract, the authors refer to "the identified control region", but the meaning of this will be unclear to readers without more context, since the LIFG/aINS has not yet been defined as a "control region".

p. 2, The authors suggest that Korsakoff's amnesia is caused by medial temporal lobe lesions, but it is most often caused by damage to the thalamus and mamillary bodies (of the hypothalamus).

p. 7, What do the authors mean when they say that long-term memory retrieval facilitates "our ability to ascribe meaning to our thoughts"? This seems to imply that thoughts are not inherently endowed with meaning.

p. 7, The authors should elaborate more on what they mean by "two ends of a continuum of memory expression".

p. 11, The TE for the MRI sequences should be reported in its actual ms value rather than as the "minimum full" setting that was selected on the GE scanner console.

Reviewer #3 (Remarks to the Author):

The authors present nice data on similar neural functional modules between episodic and semantic memory in experimental settings, focused on the involvement of different levels of cognitive control. This is a timely topic, because there has been a lot of interest in the intersection between these classically differentiated forms of memory. Overall, this is positioned to be impactful and well-cited.

I struggled somewhat with the framing of the study and manuscript.

1 -- My first concern involves how the distinction between episodic and semantic memory is presented in motivating the current work.

The authors summarize: "... This raises the possibility that the previously-reported distinctions in the neural retrieval processes for semantic and episodic memory might partially reflect quantitative differences in memory strength across the laboratory-based tasks that are commonly used to probe these two memory types. In other words, the degree of automatic re-activation of strong memory traces versus controlled retrieval processes required to access weak memories under confined experimental settings might constitute an important feature of the long-held neural distinctions made within long-term memory systems.

...Difficulties on episodic memory tasks only arise when the cue is inadequate to retrieve the relevant information – for example, when a cue is linked to multiple memory traces, generating interference, or when an episodic memory is weakly encoded as a result of little practice or exposure. However, when retrieval is successful, rich details about internal thoughts and the environment that were present at encoding can be automatically reinstated in order to meet task demands.

...Collectively, these inherent differences in the experimental tasks that probe semantic and episodic memory give rise to two alternative hypotheses on the neural mechanisms for long-term memory retrieval: (i) Classical differences in the neural engagement observed across these two memory tasks might reflect distinct retrieval processes for semantic and episodic memory; (ii) alternatively, there might be a common neurocognitive process involved in access to both semantic and episodic memory, but the experimental tasks may place varying demands on more automatic versus controlled forms of retrieval, giving rise to apparent differences in semantic and

episodic memory networks. ""

The suggestion made appears to be that some or many of the neural differences reported are due to cognitive differences in automaticity, and that these are not part of the definition of these forms of memory but instead reflect a failure for prior scholarly work to have properly equated these demands. The argument then is that if one were to properly match the subjective and objective aspects of these forms of memory more closely, then one might reveal common neural underpinnings.

The authors also appear to attribute greater potential automaticity to episodic recollection – which is surprising to me given a rich history tying recollection to control processes, some of which the authors do cite (e.g., Badre and Wagner, 2007).

The authors could be more clear in their introduction on the problem statement. What aspects of episodic and semantic memory do they consider meaningful and that should be left intact in experimental settings? What aspects have not been well controlled in prior studies – but could be in the right design without changing the nature of the memory being studied?

2) If the authors' emphasis is indeed on specifically controlling depth-of-processing between memory types, it is not clear that their design achieves that or equivalent automaticity. Their data (distributions only shown for experiment 1) indicate that episodic and semantic conditions were not equivalent overall, or in the delta between strong and weak. While this can be "regressed out" in analysis, it's not clear that such statistical approaches would wholly skirt the problem.

2 -- My second framing concern is that common neural correlates for episodic and semantic memory – despite their subjective and experiential differences highlighted by the authors – have been well-documented, and this includes IFG. The authors cite, for example, (relatively) aged documentation on this – Burianova and Grady, 2007; Burianova, McIntosh, Grady, 2010. See also "Neural correlates of metacognitive monitoring during episodic and semantic retrieval" from Elman and colleagues, 2012, which presents data highly relevant to this work. The framing of the authors also appears to touch on recent interest in personal/autobiographical semantics and the intersections between episodic and semantic memory (e.g., Interdependence of episodic and semantic memory: evidence from neuropsychology from Greenberg and Verfaellie 2010; Personal semantics: at the crossroads of semantic and episodic memory from Renoult et al., 2012; Differential Medial Temporal Lobe and Parietal Cortical Contributions to Real-world Autobiographical Episodic and Autobiographical Semantic Memory from Brown et al., 2018), some of which they cite.

Collectively, it raises concerns about the novelty – when introducing the study, the authors could better highlight what is not known or unclear from extant data, especially in cases where the neural correlates are similar to those found here. Replication is also very important! But that's not how it's presented.

3 – The data on individual differences are potentially quite impactful. However, the analyses and interpretations appear to be quite post-hoc, and thus it is difficult to see how they deepen our understanding of the episodic-semantic distinction (or similarity, in this case). The introduction does not motivate IFG-vmPFC decoupling as a predictor of performance, and the inferences on this discovery appear to rely on (first) using the tasks to identify an IFG ROI, (then – albeit in a new sample) compute connectivity differences on the same tasks for this ROI, (then) using neurosynth to verify the vmPFC cluster found has been associated in other studies with memory tasks, and (then) to compute connectivity for that vmPFC cluster as a seed region of its own. This is exploratory, descriptive, and a somewhat circular analytic approach. By defining the seed ROIs in this way, and tracing their connectivity and functional associations, it is not clear that the authors would have expected another outcome.

The second experiment, for me, would be greatly strengthened by more contextualization for the ROIs studied in the Introduction (although at this stage, that would of course itself be posthoc – but that can't be helped), and establishing a control analysis and/or scenario in which there would be an alternative outcome. For example, the authors note that some brain areas have been attributed distinctly to recollection vs semantic processing – if they were to show that these areas, in contrast to IFG have memory domain-specific connectivity profiles, or that in contrast to IFG-vmPFC there are IFG-XXX patterns that distinguish the memory types, this would tell us something about the specificity of these effects. Is IFG a general control region that is agnostic to memory type, in contrast to ___? Or perhaps it is a hub that is involved in control for different forms of memory, but it does differentially engage, as needed, with other cortical nodes that discriminate

these forms of memory.

Additionally, the authors put a fair bit of emphasis on the neural correlates being for “weakly associated” memory traces – but this doesn’t appear to be well-characterized in their individual differences analysis. By looking at the contrast of weak vs strong, the reader is left to wonder where the “movement” in the data lie across people. Could the authors unpack this a bit by characterizing different connectivity profiles in Experiment 2 according to strong vs weak performance? For example, some people presumably struggle more with both strong and weak conditions, some more with weak specifically, and some more for semantic vs episodic. I believe there’s an opportunity to learn more about what aspects of individual memory performance are tracked by IFG resting-state connectivity.

Reviewer #1 (Remarks to the Author):

This is an interesting study on a timely and important topic, namely declarative memory and similarities between semantic and episodic memory in particular. The paper was clear, well written and used an elegant design to compare semantic and episodic retrieval across different levels of association strengths (and thus retrieval demands).

We thank the reviewer for their positive feedback and insightful comments, which have considerably improved our manuscript. Below we provide point-by-point responses to the concerns raised by the reviewer.

The finding of similar control networks is somehow expected based on the authors previous publications. More generally, it is uncontroversial to classify the control network as related to executive functions (and executive control over retrieval) and thus not necessarily distinct for episodic and semantic memory, as per the authors' current findings and consistent with recent literature. The authors could cite recent works on this topic coming to similar conclusions on the similarity in the neural bases of semantic and episodic memory, including executive control over retrieval (e.g., Keane, Bousquet, Wank, Verfaellie, 2019; Renoult, Irish, Moscovitch, Rugg, 2019; Duff, Covington, Hilverman, Cohen, 2020; Solomon & Schapiro, 2020).

We thank the reviewer for bringing these publications to our attention. Indeed, based largely on this recent literature, we had strong *a priori* hypothesis on shared neural responses in the executive control of memory retrieval across the semantic and episodic systems. As such, our objective was to design an experiment that could formally test this hypothesis with the necessary "strength" manipulation across the two memory domains. This, to the best of our knowledge, is a novel experimental contribution to the converging perspectives that underline the vital role of cognitive control in memory retrieval and shared neural processes across semantic and episodic memory domains.

In line with the reviewer's request, we have now added and further underlined these references in both the introduction and discussion sections of our manuscript.

I was not totally clear with the idea presented in the introduction that typical semantic and episodic tasks are not matched in control processes (lines 87-89). The authors suppose that semantic tasks would typically require a high degree of control, but not episodic memory tasks (Line 106-108): "However, when retrieval is successful, rich details about internal thoughts and the environment that were present at encoding can be automatically re-instantiated in order to meet task demands." If words automatically activate their meanings (e.g., Neely & Kahan, 2001) then in many typical semantic tasks (e.g., sentence verification, concrete-abstract classification, living/non-living categorization, etc) little cognitive control may need to be applied; a similar point made by the authors for episodic memory.

We thank the reviewer for raising this important point and apologize for the lack of clarity in our manuscript. Our main objective in the introduction section was to highlight the fact that controlled retrieval demands can vary across both semantic and episodic tasks but that such demands are rarely if ever manipulated in a comparable fashion across these memory domains. Consequently, the contribution of cognitive control processes to retrieval networks for episodic and semantic memory remains unknown. Although some semantic tasks, such as concrete-abstract classification and living-non-living categorization can be based on

automatic retrieval of strong exemplars (please also see Figure S4 below from our supplementary information on the conjunction of *strong versus weak* contrast across the two memory domains), many semantic tasks require control to resolve competition between alternative choices or ambiguity about the target meaning, or to recover weakly-encoded features or associations (Noonan et al., 2013; Jackson, 2020). These manipulations are more difficult to include in tasks that are commonly employed to study episodic memory retrieval (e.g. item-recognition versus familiarity tasks). As such, studies that assess variable control demands in episodic memory recall remain highly scarce and limited (Badre and Wagner, 2007; Renoult et al., 2019).

In other words, what we postulated is not that “semantic tasks typically require a high degree of control, but not episodic memory tasks”, but rather that there may be a bias in the literature in the control demands of the tasks that have typically been used. Apparent differences in the neural correlates of semantic and episodic memory retrieval (typically conducted by isolated meta-analyses across the two literatures) may reflect uncontrolled differences in memory strength manipulations, especially given that more “automatic” semantic retrieval is associated with activation in regions associated with episodic memory.

Figure S1. Shared processes in the automatic retrieval of long-term memory. In addition to controlled retrieval investigated in the main study, we further interrogated any commonalities in the neural responses for the retrieval of *strong* versus *weak* associations across memory types. (A) For both semantic and episodic 3-AFC fMRI tasks, shared neural responses were observed for the retrieval of strongly versus weakly associated word-pairs. (B) A formal conjunction analysis illustrated that for the *strong > weak* contrast, the posterior cingulate/precuneal cortices (PCC/PCUN), right angular (RAG) and middle temporal gyri (MTG) showed greater activity for both memory types, which almost exclusively overlapped with the default mode network (based on the Yeo 7-Network parcellation).

We have now extensively revised our introduction section to clarify this point further. The relevant section now reads:

“Despite these long-standing distinctions in the cognitive and neural instantiations of semantic and episodic memory, emerging evidence now calls into question the extent of their separation (Renoult et al., 2019). Specifically, common cognitive processes are suggested to underlie the large overlap that is observed in the retrieval networks supporting semantic and episodic memory (Irish and Vatansever, 2020). One core process that is arguably shared across the two memory domains is cognitive control. Generally defined as a goal-directed executive system, cognitive control is postulated to allow the flexible adjustment of prepotent responses to better meet changing and often ambiguous environmental demands. For both semantic (Jefferies, 2013) and episodic (Badre and Wagner, 2007) memory, cognitive control is required when dominant memory traces are not sufficiently strong

enough to drive appropriate behaviour in an unambiguous manner (e.g. distinguishing between a bee and a wasp for their likelihood to sting). In the case of semantic memory, the automatic retrieval of strongly associated word pairs is consistently linked to activity in brain regions in the posterior parietal cortex (Whitney et al., 2012; Davey et al., 2015) that partly match the posterior medial network attributed to episodic recollection (Thakral et al., 2017). Conversely, the left inferior frontal gyrus that is commonly activated in the controlled retrieval of semantic information (Whitney et al., 2011) also shows engagement when participants are asked to retrieve weakly encoded episodic memory traces (Barredo et al., 2015). Together, this evidence highlights cognitive control as an important aspect of memory retrieval across the two domains with comparable neural instantiations (Badre and Wagner, 2007). This in turn raises the possibility that the previously reported distinctions in the neural retrieval mechanisms for semantic and episodic memory, which are often based on single studies or isolated meta-analyses conducted across the two sets of literature, might partially reflect quantitative differences in control demands across the laboratory-based tasks that commonly probe these two memory types. In other words, the degree of automatic re-activation versus controlled retrieval processes required to access memory traces under confined experimental settings might constitute an important feature of the long-held distinctions made between the neural retrieval networks of the two long-term memory systems.”

Line 297-299 (see also introduction lines 80-82). I did not really understand why the authors compared the semantic control network to the posterior memory network. Would it not make more sense to compare the semantic retrieval network to the episodic retrieval network (that some may call posterior memory network) that recent literature mentioned above show had clear overlap?

We thank the reviewer for highlighting this important point. Our aim in describing these two networks was to illustrate exemplars for the neural correlates of the two memory systems that best reflect the current state of the literature across these two memory domains. “The general semantic network” (or semantic retrieval network) was initially defined by the meta-analysis conducted by Binder and colleagues in 2009, based on a careful consideration of over 200 studies. Although this meta-analysis focused on a wide array of semantic tasks, it largely disregarded tasks with high control demands. The meta-analysis by Noonan and colleagues in 2013 (and more recently by Jackson (2020)) on the other hand included these manipulations and highlighted brain areas important for semantic control. As such, we believe that this later definition was more representative of overall “semantic retrieval”.

Although we completely appreciate the reviewer’s comment on the large spatial overlap across these two memory retrieval networks, which we also recently highlighted (Irish and Vatansever, 2020), there remains a level of discrepancy that is also evident in our topic-based meta-analytic search across the Neurosynth database (Figure 2B). Nevertheless, as per the reviewer’s request, we now updated the referenced sections in order to better reflect previous definitions of semantic and episodic retrieval networks in the main manuscript. The relevant section now reads:

“Important in the active selection and manipulation of conceptual information, both “general semantic” (Binder et al., 2009) and “semantic control” (Noonan et al., 2013; Jackson, 2020) networks have been identified through meta-analytic approaches, revealing regions that span across the inferior frontal cortex, posterior middle temporal gyri, inferior parietal sulcus as well as regions in the cortical midline.”

One of the main conclusions of the paper is that differences in neural correlates usually reported between semantic and episodic memory would be due to differences in cognitive control (e.g., lines 140-142: “Thus, classical distinctions in the semantic and episodic retrieval networks in the literature may reflect differences in the cognitive control requirements of commonly used long-term memory tasks”; see also discussion lines 291-294). It was not clear to me how this would work. As mentioned above, quite a few investigators came to the conclusion recently that semantic and episodic retrieval networks are very similar. Typical differences in neural correlates between semantic and episodic memory may not be easily interpreted in term of cognitive control. For instance, quite a few investigators would think (consistent with literature presented in this manuscript) that the hippocampus has a clear association with episodic memory (and perhaps less so for semantic memory) and that the anterior temporal lobe has a clear association with semantic memory (and perhaps less so with episodic memory): how do these differences can be accounted in term of the control network?

We thank the reviewer for highlighting this important point and apologize for the confusion. In the statement highlighted by the reviewer, our focus was on the classical distinctions made across the “retrieval” networks, particularly greater involvement of the left inferior frontal and posterior middle temporal regions in the retrieval of semantic memory – areas which are largely absent from traditional episodic memory retrieval networks. Based on the evidence provided, it is these apparent differences that we largely attribute to differences in memory strength, and thus cognitive control demands. Nevertheless, such a conclusion does not eliminate the possibility that there might be spatial differences in the “storage, representation and/or encoding” of memory across these two domains. As the reviewer highlighted, a large body of neuropsychological and clinical literature suggests that the ATL is a neocortical semantic storage hub for heteromodal/abstract knowledge, while the MTL plays a more confined role in the acquisition (or reconstruction) of spatiotemporal memory traces. Nonetheless, based on the information we presented, we envisage large interdependence and overlap in the context-dependent retrieval of memory across the two domains in order to flexibly guide goal-directed behavior, which we believe is in line with the recent perspectives highlighted by the reviewer.

We have now altered our discussion section in order to clarify this distinction as follows:

“This finding is consistent with the theoretical insight that while there may be complementary representational systems – with anterior temporal lobe representing commonalities across experiences and hippocampus supporting pattern separation – retrieval pathways for semantic and episodic memory may be partly overlapping, such that stronger and more coherent experiences across both types of memory may be supported by regions within the default mode network, while weaker and less coherent experiences recruit control regions (Noonan et al., 2013; Vatansever et al., 2017a; Irish and Vatansever, 2020). Hence, greater separation of the intrinsic connectivity linked to these component processes may reflect greater control of memory over automatic aspects of cognition (Vatansever et al., 2017b; Vatansever et al., 2017a).”

The point made line 93-103 that semantic tasks typically investigate knowledge learned over many years, while lab-based episodic memory tasks investigate recently learned associations is a fair one, but is not valid for the rich literature of autobiographical memory studies, where one often probes semantic and episodic details for the same memories. The core recollection network observed by these tasks is very similar (or identical) to that observed in lab-based studies (e.g., Moscovitch et al., 2016).

We thank the reviewer for bringing this literature to our attention. We have now modified this section in the introduction in order to incorporate the exception of autobiographical memory retrieval in this context.

Reviewer #2 (Remarks to the Author):

Vatanev and colleagues conducted two well-powered fMRI studies investigating the possibility that both episodic and semantic retrieval tasks rely on a common cognitive control process. In the first study, subjects (N=46) were each scanned twice as they performed two carefully matched tasks of episodic and semantic retrieval. As with many prior studies, the authors found some regions that were more active for semantic retrieval and others more active during episodic retrieval. But the core analyses focused on contrasts between trials with weak vs. strong memories (either based on pre-experimental associative strength [semantic strength] or number of study trials [episodic strength]). The key finding is that one brain region – a large swath of left inferior frontal gyrus and anterior insula – showed increased activity for weak memories during BOTH episodic and semantic retrieval. The authors interpret this overlap as indicating that the region makes a common contribution to mnemonic control processes, and that its function should not be attributed to a particular memory system. The authors then use this region as a seed for a whole-brain resting state functional connectivity analysis conducted on an independent fMRI dataset (N=140). Their main finding is that the strength of this region's intrinsic anti-correlation with the ventromedial prefrontal cortex was correlated with individual differences in performance on both the semantic and episodic memory tasks.

Understanding the similarities and differences between episodic and semantic memory is a hot topic in the field right now, and this paper's intriguing set of results will be read with interest by many memory researchers. I was impressed with the effort that went into collecting such large fMRI datasets (including one experiment that involved sessions across three separate days), and I thought the perceptually matched episodic and semantic retrieval were cleverly designed. Generally speaking, the fMRI data analyses were thoughtfully executed and the well-crafted figures provide informative visualizations of the findings. That said, I have some concerns about the interpretability of the findings, and there are issues with a few of the analyses that I think should be addressed.

We are delighted to receive this positive feedback on our study design, sample size and analyses. We also thank the reviewer for their helpful comments which have considerably improved our manuscript. Below we provide point-by-point detailed responses and outline the necessary changes that were made to the manuscript.

1. The core finding of the paper seems to be that the left inferior frontal gyrus (LIFG) is involved in both episodic and semantic retrieval, with the implication that its role has more to do with cognitive control processes that facilitate access to weakly associated memories than with the recovery of actual episodic or semantic content per se. However, an alternative interpretation of the results is that this region's contributions are heavily semantic in nature, and that many episodic memory tasks (especially those like the present study that involve building memorable associations between seemingly unrelated words) have a strong conceptual processing component. Indeed, a recent review by Renault, Irish, Moscovitch, Rugg (TiCS, 2020) states that: "consistent with the notion that the left inferior prefrontal cortex supports semantic control processes... it has been reported that the region is more active during episodic memory tests that place high, as opposed to low, demands on the processing of retrieval cues."

In the context of the present experiment, the strong episodic trials assess memory for unrelated word pairs that had been studied six times. If a subject had six opportunities to learn the pairing of APPLE-FLUTE, they probably will have built up a richly elaborated

semantic connection between these two concepts, such as visualizing someone play a flute under an apple tree). Word pairs that were studied only twice (weak episodic trials) probably also involved the generation of semantic linkages between the referents, but these associations may have been more tenuously constructed and thus will place greater demands on semantic processing during retrieval. The episodic task was trivially easy for subjects, yielding 99% correct on strong trials and 98% correct on weak trials, so subjects were almost always able to successfully recall which of the three probe words had been paired with the target word. The extra time that participants took to respond on weak trials may have been used to mentally simulate a semantic connection between the weakly linked words and evaluate whether that connection felt like one they had thought of the prior day. My general point here is that presence of overlapping LIFG activation in the weak > strong contrasts for the episodic and semantic tasks does not rule out the possibility that differential semantic processing is responsible for both contrasts. Perhaps the authors could partially rule this out by running a task-related functional connectivity analysis and showing a differential profile of connectivity of the LIFG depending on whether episodic vs. semantic knowledge needed to be accessed. But even such a demonstration would not adjudicate between the authors' proposed strength-based account and an account that posits that the overlap is due to common demands for semantic processing on the weakly associated trials of both tasks.

We thank the reviewer for raising this important point, which refers back to the question of how separable these two theoretically (and functionally) different memory systems actually are. To clarify, our aim was not to design two separate tasks that would “perfectly isolate” the semantic and episodic memory systems. In fact, we do not believe that it is possible to achieve such isolation due exactly to the reasons outlined by the reviewer. There is converging evidence from both clinical and cognitive perspectives which now indicate the interdependence of these two systems. Episodic tasks (specifically those that require autobiographical memory retrieval) require access to semantic stores, and vice versa (e.g. personal semantics). This was also underlined by the excellent review paper highlighted by the reviewer (Renoult et al., 2019).

Our point is that the only manner in which we can identify common and separable neural correlates of these two systems is by using a common manipulation across the two domains, which in this instance was memory strength. For this manipulation to be comparable across the two memory systems, we aimed to achieve a level of experimental similarity on the basic visual presentation of stimuli. For the episodic task, although it was not possible for us to facilitate the encoding of ecologically-valid autobiographical memory traces, we used a task design that was comparable to those that previously aimed to assess episodic memory retrieval in laboratory settings (Ritchey and Cooper, 2020). Although the author is correct in suggesting that we cannot eliminate the potential influence of “semantic memory” on episodic retrieval, we believe that the strength manipulation we used was specific to the retrieval of episodic memory traces with no apparent difference in semantic concepts. This is because:

- 1) The design we employed is similar to the original formulation of semantic and episodic memory as two distinct functional domains by Endel Tulving (e.g. episodic memory as assessed by remembering whether a word was part of a list of words presented previously) (Tulving et al., 1972).
- 2) Conceptual representations are usually formed over a lifetime and the level of short-term training provided would not be sufficient enough to establish such meaning-based links to be formed.

- 3) The study design required participants to learn the associations, which were then tested at least 24 hours after encoding. Such a manipulation is commonly employed in experimental tasks of episodic memory (e.g. item-recognition versus familiarity tasks).
- 4) As per the reviewer's formulation, "mental simulations" and "visualizing someone playing the flute under a tree" are processes that would generally be attributed to the episodic and not the semantic system (Renoult et al., 2019).

Nevertheless, as per the reviewer's request, we now explicitly express this concern in our limitations section as follows:

"Notable limitations should be considered when interpreting the results of our study. First, although the retrieval paradigms in our Experiment 1 were designed in line with the current literature and in a manner that would categorically separate semantic and episodic memory, perfect isolation of memory retrieval across the two domains could not be guaranteed. Conceptual information could have been retrieved during the episodic memory task. Conversely, rich autobiographical and episodic memory could have been re-activated during the performance of the semantic memory task. Hence, a common strength manipulation across both paradigms was necessary to deduce the effects of control demands that are largely independent of domain categorization. Nevertheless, further research with variable paradigms and stimulus modalities will be required to assess the reliability of our results to operational categorization of different memory types."

2. In Fig 3E, the authors show a scatterplot of the correlation between participants' LIFG activation level during weak trials of the episodic and semantic tasks. The take-home point of this analysis is not entirely clear, as little is said about why it was done or what it means. But the analysis itself seems potentially problematic. Rather than computing weak > strong activation difference scores for each subject and then correlating those across tasks, the authors are correlating the single condition parameter estimates for weak trials (vs. the implicit fixation baseline). Thus subjects with more robust task-related BOLD responses (perhaps attributable to better neurovascular coupling, more alertness, more caffeine, or less head movement, etc.) will tend to show higher activation parameter estimates in both tasks. To rule this out and show that the effect is specific to trials with weakly associated stimuli, the authors should demonstrate that no such correlation exists between the episodic and semantic "control task trials", or between strong episodic and strong semantic trials.

We thank the reviewer for highlighting this important point and apologize for the lack of clarity associated with this analysis. The aim of this analysis was to further highlight the contribution of this conjunction region to both memory types across participants. Although the main analysis allows us to make inferences at the group level, it is possible that the observed effect varies at the individual level across the two tasks. For that purpose, we wanted to test whether greater responses in the identified region during the retrieval of weakly associated word pairs in the semantic task was also linked to greater responses to the retrieval of weakly encoded word pairs in the episodic task. However, we fully take on board the suggestions made by the reviewer in order to further strengthen these results and have now carried out the additional analysis requested by the reviewer.

The results indicated that there was no significant correlation in the parameter estimates of this conjunction region obtained from the control trials (in comparison to implicit baseline) across semantic and episodic memory retrieval tasks (partial $r_p = .073$, $p = .32$, corrected for

age and gender). In order to streamline this analysis further and incorporate the common control trials in the main manuscript, we now report on the significant correlation in parameter estimates in *weak > control* contrast across semantic and episodic memory tasks (partial $r_p = .27$, $p = .036$, corrected for age and gender). No significant correlation was observed for the same analysis in the *strong > control* contrast across the two memory tasks (partial $r_p = .11$, $p = .24$, corrected for age and gender).

Collectively, this evidence points to specific responses of this region to weakly encoded semantic and episodic memory retrieval. We have now added this information both to the main manuscript and the supplementary information and altered Figure 3 accordingly. The relevant section now reads:

“Moreover, in addition to the observed main effects at the group level, individual variability in brain activity also depicted significant correlations across the two memory types. Specifically, activity differences in the retrieval of weakly associated word pairs in comparison to control trials was significantly correlated across the semantic and episodic memory retrieval paradigms (partial $r_p = .27$, $p = .036$, corrected for age and gender) (Fig. 3E). However, no significant correlation was observed in activity differences between the strong versus control trials (partial $r_p = .11$, $p = .24$, corrected for age and gender). Overall, these results indicate shared neural responses during the retrieval of weakly associated word pairs across the two memory types. As such, our findings suggest that at least part of the memory type differences observed in our initial analysis and in other prior investigations might arise due to underlying differences in memory strength, and thus the level of control demands required to access long-term memory traces.”

Figure 3. Shared brain activity patterns associated with controlled retrieval across long-term memory types. (A) For both semantic and episodic 3-AFC fMRI tasks, shared neural responses were observed in the retrieval of weakly versus strongly associated word-pairs. (B) A formal conjunction analysis illustrated that for the *weak > strong* contrast, left inferior frontal gyrus (LIFG) and anterior insular cortex (aINS) showed greater activity for both memory types, which largely overlapped with the fronto-parietal, salience/ventral attention and default mode networks (based on the Yeo 7-Network parcellation (Yeo et al., 2011)). (C-D) The peristimulus time plots (for illustration purposes only) showed sustained activity differences between weak versus strong trials (in comparison to implicit baseline) that spanned 1-3 repetition times (TRs) across all participants for both memory types. Curved lines represent the mean, while shaded areas reflect 95% confidence intervals. (E) LIFG/aINS activity differences during weak versus

control trials between semantic and episodic memory tasks were significantly correlated across participants (partial $r_p = .29$, $p = .036$, correcting for age and gender). Straight line represents the best fit, while shaded areas illustrate 95% confidence intervals. No such correlation was observed for activity differences between strong versus control trials across the two memory types (partial $r_p = .11$, $p = .24$, correcting for age and gender).

3. The authors show hemodynamic timecourse plots in Fig 3C and 3D to illustrate “a clear separation in brain activity associated with the weak versus strong association trials for both memory types”, but this is circular claim, since the region-of-interest was defined precisely on the virtue of it showing this weak > strong effect. If the timecourse plots have some additional informational value that the authors believe is worth conveying to readers, they should have a clearly stated circularity warning (e.g., “shown for illustrative purposes only”).

We thank the reviewer for highlighting this important point. We completely understand the reviewer’s concern about circularity. Our aim was not to conduct additional analyses, but was rather to show more in-depth information on the observed effect across participants and across time. Peristimulus timecourse plots are commonly used in event-related task designs and are useful in detecting whether a region shows increased amplitude of instantaneous activity or one that is sustained across the task trials.

We have now altered the corresponding results section of our manuscript and added a warning in the figure caption indicating that the plots are for illustrative purposes only.

4. I have a similar concern about inferential circularity that pertains to the scatterplot shown in Fig 4A. The authors conducted a whole-brain seed-based connectivity analysis and found a significant connectivity-behavior relationship in the vmPFC. They then show us a scatterplot depiction of this correlation from that very region, which was defined entirely on the basis of it showing this effect. Unless I am missing something about the way that this scatterplot figure was derived, it is a misleading figure that provides little informational value and should be removed.

We thank the reviewer for raising this important point. Again, our aim here was not to conduct an independent analysis, but was rather to provide an in-depth illustration of the identified relationship. The scatter plot in this context is useful in showing the distribution of values across the two axes and in helping the reader deduce the direction of the observed relationship, which is not readily available on brain surface images. Furthermore, categorizing this connectivity and behavioral performance relationship across the two memory domains is again helpful to visualize their respective contributions. As such, we believe that keeping this figure will allow the readers to better understand our results. Nevertheless, as per the reviewer’s request for the PSTH plots, we have now added a circularity warning, highlighting that the plot is for illustrative purposes only.

5. More broadly, I struggled to understand what was learned about the nature of episodic and semantic memory by doing this analysis of individual differences in resting state connectivity. The authors show us that the intrinsic anti-correlation between the LIFG/aINS and the vmPFC is stronger in those individuals with better memory performance. But how do we even know that this effect has anything to do with memory ability per se? Because there is no non-memory control task, it is possible that individuals who are better performers in general (perhaps because of higher motivation, attentional focus, decision-making, intelligence, and/or reading ability) would also be the ones with the highest anti-correlation between these two regions. Even if the effect was proven to be specific to memory ability, the authors should say more about what they think this

effect means (especially in light of the fact that a big portion of their LIFG/aINS cluster is designated as a DMN region).

We thank the reviewer for highlighting this important point and apologize for the lack of clarity associated with this analysis. Our general aim with this resting state analysis was to test whether individual trait-level variability in the functional interactions of the identified region was associated with performance differences in the retrieval of weakly associated memory traces across both semantic and episodic memory domains (while correcting for the potential influence of age, gender and in-scanner head motion).

Following the reviewer's suggestion, we conducted additional analyses to ascertain whether the observed link could be explained by individual differences in fluid intelligence or selective attention / inhibitory control. For that purpose, we employed two sets of additional measurements that were collected from this participant cohort. While fluid intelligence was measured via the number of correct trials in Raven's Advanced Progressive Matrices (Raven and Raven, 2003), reaction time to incongruent trials in the Flanker task was used to assess selective attention / inhibitory control (Eriksen and Eriksen, 1974). Neither individual differences in fluid intelligence (partial $r_p = -.010$, $p = .45$), nor selective attention / inhibitory control (partial $r_p = .052$, $p = .27$) showed a significant correlation with individual variation across the functional connectivity between LIFG/aINS and vmPFC (corrected for age, gender and in-scanner head motion). We now include these findings in the manuscript and the supplementary materials as follows:

"Moreover, individual variation in the connectivity strength of this neural link was neither related to fluid intelligence (partial $r_p = -.010$, $p = .45$), nor selective attention / inhibitory control (partial $r_p = .052$, $p = .27$) as measured via accuracy-based indices in the Raven's advanced progressive matrices (Raven and Raven, 2003) and reaction time to incongruent trials in the Flanker (Eriksen and Eriksen, 1974) tasks, respectively (Supplementary Information S2)."

The functional connection we identified in this study is consistent with our recent publication in which we assessed the link between connectivity across regions commonly associated with semantic processing and performance in a battery of semantic tasks using canonical correlation analyses (Vatansever et al., 2017b). This work showed better performance across semantic tasks that required the controlled retrieval of weak conceptual knowledge in participants with stronger anti-correlation between the left inferior frontal gyrus (part of the fronto-parietal network) and angular gyrus (part of the default mode network). The findings are also in line with our previous study showing that disruption of the left inferior frontal gyrus with transcranial magnetic stimulation (TMS) selectively impairs the retrieval of weakly associated word pairs, but not strongly associated word pairs or a perceptual task (Whitney et al., 2011). In parallel, in our current study we found that the connectivity of LIFG/aINS (largely part of the fronto-parietal network) to the vmPFC (largely part of the default mode network) was commonly linked to performance across both semantic and episodic memory tasks.

Recent evidence indicates the important role that the vmPFC might play in in schema-based (or automatic) retrieval of memory (Gilboa and Marlatte, 2017). Collectively, this evidence potentially indicates the need to separate default mode from control networks in order to exert better control over retrieval and efficiently retrieve weak associations. Further research will be required to deduce the causal influence of the identified region in the

controlled retrieval of memory across participants. We have now added this interpretation to the discussion section of our manuscript. The relevant section reads as follows:

“This finding is consistent with the theoretical insight that while there may be complementary representational systems – with anterior temporal lobe representing commonalities across experiences and hippocampus supporting pattern separation – retrieval pathways for semantic and episodic memory may be partly overlapping, such that stronger and more coherent experiences across both types of memory may be supported by regions within the default mode network, while weaker and less coherent experiences recruit control regions (Noonan et al., 2013; Vatansever et al., 2017a; Irish and Vatansever, 2020). Hence, greater separation of the intrinsic connectivity linked to these component processes may reflect greater control of memory over automatic aspects of cognition (Vatansever et al., 2017b; Vatansever et al., 2017a). To this end, emerging evidence now highlights the important role played by the vmPFC in schema-mediated memory retrieval (Gilboa and Marlatte, 2017), in contrast to the common attribution of the left inferior frontal gyrus to controlled retrieval processes (Badre and Wagner, 2007). Nevertheless, further task-based studies with causal methods (e.g. transcranial magnetic stimulation) and finer time resolution (e.g. magnetoencephalography) will be required to understand the directional exchange of information in the service of controlled memory retrieval.”

6. The authors feature the vmPFC as being the only region that showed a significant relationship between its connectivity with the LIFG/aINS and behavioral performance on the two tasks. However, Fig 4B shows an unthresholded view of the same map, and there seem to be comparably robust connectivity-behavior relationships in number of other default mode network regions, including bilateral angular gyrus, posterior cingulate cortex, and superior frontal gyrus. Did none of those other effects achieve significance?

The author is correct in their interpretation of our results. Although the unthresholded map does in fact show other brain regions with high correlation, the results did not reach statistical significance in this analysis.

6. The use of Neurosynth to label the relative network membership composition of activation clusters or whole brain maps is a cool addition to the paper, but I would like the authors to elaborate more on what is learned by knowing these relative network membership compositions for each analysis. For instance, the major focus of this paper is a large LIFG/aINS cluster that seems to span across three different functional networks. Do the authors suspect that different components of this large cluster contribute in different ways, depending on their network membership?

We thank the reviewer for raising this important point. Our aim in labeling the activation clusters based on the Yeo7 network parcellation scheme was to standardize the nomenclature that was used to describe different regions across different analyses, and to improve interpretation of our task-based findings in relation to the resting state analysis. To give an example, the “multiple-demand network” that is commonly identified in difficult task-based fMRI contexts is also composed of a combination of “intrinsic brain networks” measured at rest. It is highly plausible that different sections of the identified cluster have different functional roles in memory retrieval or rather reflect “convergence zones” where dissociable brain networks interact for the control of memory retrieval. Although this was beyond the scope of our investigation, it will be exciting and necessary to test these hypotheses in future studies.

7. Although I found it visually appealing, it is ultimately misleading to print the Neurosynth word cloud terms on top a brain in Fig 4B, since it gives a false impression of region-to-function mapping.

We thank the reviewer for highlighting this point. As per the reviewer's request, we have now removed the background brain image on this figure.

Minor comments:

Was the fMRI data analysis limited to correct trials only?

We confirm that the analysis was limited to correct trials only, which we now further clarify in the methods section. This was achieved by using a trial weight of 0 and 1 in the design matrix for the general linear model for incorrect and correct trials, respectively.

What was the cluster extent threshold requirement for the whole brain analyses?

The cluster forming threshold for the whole brain analyses was $z > 2.6$. We now further highlight this in our manuscript.

How did the authors get such fine-grained temporal resolution in their timecourse plots when the TR was 3 sec?

The time course plots represent responses averaged across events for each participant. Given the variable jitter that was employed between events, the resulting points on the plot appear to have greater temporal resolution than repetition time. As suggested in the relevant literature, we fit a smoothing spline curve using MATLAB's "fit" function to the event and beta activity data, which was then used to interpolate and match the time courses across participants for each TR. We have now clarified this further in the methods section of our manuscript and on the corresponding figure.

p. 1, In the abstract, the authors refer to "the identified control region", but the meaning of this will be unclear to readers without more context, since the LIFG/aINS has not yet been defined as a "control region".

We have now altered the abstract to correct this issue.

p. 2, The authors suggest that Korsakoff's amnesia is caused by medial temporal lobe lesions, but it is most often caused by damage to the thalamus and mamillary bodies (of the hypothalamus).

We apologize for this oversight and thank the reviewer for raising this important point. We have now removed this example from the main manuscript.

p. 7, What do the authors mean when they say that long-term memory retrieval facilitates "our ability to ascribe meaning to our thoughts"? This seems to imply that thoughts are not inherently endowed with meaning.

We have now rephrased this sentence in order to improve its clarity.

p. 7, The authors should elaborate more on what they mean by “two ends of a continuum of memory expression”.

Our aim here was to highlight that variations along dimensions of shared attributes within long-term memory traces (e.g. memory strength, abstraction etc.) may potentially underlie some of the neural differences observed across semantic and episodic domains. We now provide further elaboration on this point in our discussion section.

p. 11, The TE for the MRI sequences should be reported in its actual ms value rather than as the “minimum full” setting that was selected on the GE scanner console.

We apologize for this oversight and thank the reviewer for bringing it to our attention. We now report the 18.9 ms TE value for the EPI sequence and 3 ms TE for the T1 sequence in the methods section of our manuscript.

Reviewer #3 (Remarks to the Author):

The authors present nice data on similar neural functional modules between episodic and semantic memory in experimental settings, focused on the involvement of different levels of cognitive control. This is a timely topic, because there has been a lot of interest in the intersection between these classically differentiated forms of memory. Overall, this is positioned to be impactful and well-cited.

We thank the reviewer for their positive feedback on our study and for the insightful comments which have considerably improved our manuscript. Below we outline point-by-point responses to the reviewer's concerns and have altered the manuscript accordingly.

I struggled somewhat with the framing of the study and manuscript.

1 -- My first concern involves how the distinction between episodic and semantic memory is presented in motivating the current work.

The authors summarize: "... This raises the possibility that the previously-reported distinctions in the neural retrieval processes for semantic and episodic memory might partially reflect quantitative differences in memory strength across the laboratory-based tasks that are commonly used to probe these two memory types. In other words, the degree of automatic re-activation of strong memory traces versus controlled retrieval processes required to access weak memories under confined experimental settings might constitute an important feature of the long-held neural distinctions made within long-term memory systems.

...Difficulties on episodic memory tasks only arise when the cue is inadequate to retrieve the relevant information – for example, when a cue is linked to multiple memory traces, generating interference, or when an episodic memory is weakly encoded as a result of little practice or exposure. However, when retrieval is successful, rich details about internal thoughts and the environment that were present at encoding can be automatically reinstated in order to meet task demands.

...Collectively, these inherent differences in the experimental tasks that probe semantic and episodic memory give rise to two alternative hypotheses on the neural mechanisms for long-term memory retrieval: (i) Classical differences in the neural engagement observed across these two memory tasks might reflect distinct retrieval processes for semantic and episodic memory; (ii) alternatively, there might be a common neurocognitive process involved in access to both semantic and episodic memory, but the experimental tasks may place varying demands on more automatic versus controlled forms of retrieval, giving rise to apparent differences in semantic and episodic memory networks."

The suggestion made appears to be that some or many of the neural differences reported are due to cognitive differences in automaticity, and that these are not part of the definition of these forms of memory but instead reflect a failure for prior scholarly work to have properly equated these demands. The argument then is that if one were to properly match the subjective and objective aspects of these forms of memory more closely, then one might reveal common neural underpinnings.

The authors also appear to attribute greater potential automaticity to episodic recollection – which is surprising to me given a rich history tying recollection to control processes, some of which the authors do cite (e.g., Badre and Wagner, 2007). The authors could be more clear in their introduction on the problem statement. What aspects of episodic and semantic memory do they consider meaningful and that should be left intact in experimental settings? What aspects have not been well controlled in prior studies – but could be in the right design without changing the nature of the memory being studied?

We thank the reviewer for this insightful comment and apologize for the lack of clarity regarding this point in the introduction section of our manuscript. To clarify, our objective was not to match the “subjective and objective aspects of these forms of memory” and then to “reveal common neural underpinnings” across the two memory types. In fact, we do not believe that it is possible to achieve such matching given latent differences in the manner in which these two memory systems interact with cognition and are manifested in neural responses.

Conversely, our suggestion is that the only way to assess commonalities and differences in the neural correlates of these two memory systems is to introduce a common manipulation across the two memory types. The manipulation used here was “memory strength”, since this is thought to affect the control demands of memory tasks. As highlighted by the reviewer, memory strength and thus cognitive control processes that are required to deal with retrieval demands, have been highlighted in both the semantic and episodic memory literatures. The major problem is that this manipulation has been employed in two separate streams of research focused on these two memory types, with no direct comparisons available to show how this strength manipulation affects different aspects of memory retrieval in common or distinct ways. While strength manipulations are relatively common for semantic tasks (e.g. semantic associations, general or specific item categorization), similar manipulations are rarely employed in episodic tasks. This may either be due to the ease with which one can manipulate strength in semantic as compared to episodic tasks or may be rooted in quantitative differences in the manner in which humans experience and process these two memory types. Collectively, this raises the possibility that hidden differences in memory strength in specific paradigms may contribute to apparent differences in neural activity that could then be misinterpreted as differences in domain (i.e. as either semantic or episodic memory). In other words, differences in experimental approaches across the semantic and episodic memory literatures may give rise to biases when meta-analyses identify the “neural correlates of semantic and episodic memory”. This is further highlighted by the topic-based meta-analysis we present in Figure 2B.

As per the reviewer’s request, we have now extensively altered the introduction section of our manuscript in order to improve upon this point. The relevant section now reads:

“Despite these long-standing distinctions in the cognitive and neural instantiations of semantic and episodic memory, emerging evidence now calls into question the extent of their separation (Renoult et al., 2019). Specifically, common cognitive processes are suggested to underlie the large overlap that is observed in the retrieval networks supporting semantic and episodic memory (Irish and Vatansever, 2020). One core process that is arguably shared across the two memory domains is cognitive control. Generally defined as a goal-directed executive system, cognitive control is postulated to allow the flexible adjustment of prepotent responses to better meet changing and often ambiguous environmental demands. For both semantic (Jefferies, 2013) and episodic (Badre and Wagner, 2007) memory,

cognitive control is required when dominant memory traces are not sufficiently strong enough to drive appropriate behaviour in an unambiguous manner (e.g. distinguishing between a bee and a wasp for their likelihood to sting). In the case of semantic memory, the automatic retrieval of strongly associated word pairs is consistently linked to activity in brain regions in the posterior parietal cortex (Whitney et al., 2012; Davey et al., 2015) that partly match the posterior medial network attributed to episodic recollection (Thakral et al., 2017). Conversely, the left inferior frontal gyrus that is commonly activated in the controlled retrieval of semantic information (Whitney et al., 2011) also shows engagement when participants are asked to retrieve weakly encoded episodic memory traces (Barredo et al., 2015). Together, this evidence highlights cognitive control as an important aspect of memory retrieval across the two domains with comparable neural instantiations (Badre and Wagner, 2007). This in turn raises the possibility that the previously reported distinctions in the neural retrieval mechanisms for semantic and episodic memory, which are often based on single studies or isolated meta-analyses conducted across the two sets of literature, might partially reflect quantitative differences in control demands across the laboratory-based tasks that commonly probe these two memory types. In other words, the degree of automatic re-activation versus controlled retrieval processes required to access memory traces under confined experimental settings might constitute an important feature of the long-held distinctions made between the neural retrieval networks of the two long-term memory systems.”

2) If the authors’ emphasis is indeed on specifically controlling depth-of-processing between memory types, it is not clear that their design achieves that or equivalent automaticity. Their data (distributions only shown for experiment 1) indicate that episodic and semantic conditions were not equivalent overall, or in the delta between strong and weak. While this can be “regressed out” in analysis, it’s not clear that such statistical approaches would wholly skirt the problem.

Our major goal in this study was to introduce a strength manipulation across the two memory types in order to test the possibility for systematic similarities in the retrieval of semantic and episodic memory that emerges when declarative memories need to be retrieved in a particular context and goal-dependent manner. Although the reviewer is correct that the overall levels of difficulty were not conserved across tasks, the neural response of interest reflects a similar change when participants retrieve a memory with a weaker trace, which was common across domains as identified through a formal conjunction. In this context, while tasks which are better matched for memory demands could have identified greater similarity, the logic of our study allows us to conclude that the inferior frontal gyrus and anterior insular cortex are important when memories are harder to retrieve in both the semantic and episodic domains. To make this issue fully transparent, we have modified the discussion section of our manuscript and added the following limitations:

“Second, behavioural results indicate task difficulty differences across the two experimental paradigms. Although this nuisance factor was taken into account during data analysis, further studies with greater balance in task difficulty across the two memory domains may identify even more extensive overlap in the neural mechanisms of controlled long-term memory retrieval.”

2 -- My second framing concern is that common neural correlates for episodic and semantic memory – despite their subjective and experiential differences highlighted by the authors – have been well-documented, and this includes IFG. The authors cite, for example, (relatively) aged documentation on this – Burianova and Grady, 2007; Burianova,

McIntosh, Grady, 2010. See also “Neural correlates of metacognitive monitoring during episodic and semantic retrieval” from Elman and colleagues, 2012, which presents data highly relevant to this work. The framing of the authors also appears to touch on recent interest in personal/autobiographical semantics and the intersections between episodic and semantic memory (e.g., Interdependence of episodic and semantic memory: evidence from neuropsychology from Greenberg and Verfaellie 2010; Personal semantics: at the crossroads of semantic and episodic memory from Renoult et al., 2012; Differential Medial Temporal Lobe and Parietal Cortical Contributions to Real-world Autobiographical Episodic and Autobiographical Semantic Memory from Brown et al., 2018), some of which they cite. Collectively, it raises concerns about the novelty – when introducing the study, the authors could better highlight what is not known or unclear from extant data, especially in cases where the neural correlates are similar to those found here. Replication is also very important! But that’s not how it’s presented.

We thank the reviewer for helping us focus the novelty of our study and for referring us to this highly relevant literature. The studies described by the reviewer were an important source of motivation for our study as they highlighted: i) the interdependence of these two memory types in both clinical neuropsychology and cognitive studies and ii) shared neural responses in their brain-based instantiations. However, to the best of our knowledge, no existing study has directly compared the neural correlates of these two memory types together with a similar manipulation of memory strength and thus allowing the effects of retrieval demands to be understood in the same group of participants in both memory domains. In order to highlight the specific features of our design that make this study novel we have reworked the introduction section to include the literature highlighted by the reviewer and to more accurately contextualize why we conducted the study and what our results mean for our understanding of memory.

3 – The data on individual differences are potentially quite impactful. However, the analyses and interpretations appear to be quite post-hoc, and thus it is difficult to see how they deepen our understanding of the episodic-semantic distinction (or similarity, in this case). The introduction does not motivate IFG-vmPFC decoupling as a predictor of performance, and the inferences on this discovery appear to rely on (first) using the tasks to identify an IFG ROI, (then – albeit in a new sample) compute connectivity differences on the same tasks for this ROI, (then) using neurosynth to verify the vmPFC cluster found has been associated in other studies with memory tasks, and (then) to compute connectivity for that vmPFC cluster as a seed region of its own. This is exploratory, descriptive, and a somewhat circular analytic approach. By defining the seed ROIs in this way, and tracing their connectivity and functional associations, it is not clear that the authors would have expected another outcome.

We thank the reviewer for these insightful comments on our second experiment. To clarify, our objective in this individual difference analysis was to provide complementary evidence for the main findings of Experiment 1, which is based on task-based activation differences across the whole group of participants. For that purpose, we investigated how the connectivity of the LIFG/aINS region at rest was related to performance on memory retrieval in a separate set of participants. This investigation allowed the assessment of how brain connectivity at rest is associated with better retrieval and thus complements Experiment 1, which highlights shared neural patterns observed when participants remember information that is harder to retrieve. We did not anticipate the specific features of this analysis in our introduction because this analysis was based on the results of Experiment 1. As such, we did not want to give the impression that our data-driven analysis was in fact hypothesis-driven.

Nevertheless, although the results of Experiment 2 are dependent on Experiment 1, this does not make the analysis circular. In fact, by finding commonalities across the two memory tasks we can gain confidence that not only the IFG/aINS is active when people remember weak memories it also is embedded in a functional architecture that predicts better performance on semantic and episodic memory tasks. This is largely in line with prior resting state investigations that aimed to unearth the relationship between intrinsic brain connectivity profiles and various forms of cognitive aptitude, for example (Finn et al., 2015).

In line with the reviewer's request, we have now explicitly addressed this concern in our discussion section as follows:

“Finally, a data-driven analysis approach was employed in our Experiment 2 in which we selected a region of interest based on the shared activity patterns observed in our Experiment 1. Despite providing complementary findings across the two experiments, this approach was limited in identifying regions beyond the left inferior frontal gyrus and anterior insular cortex that may also relate to behavioural performance. Future studies that employ whole-brain methods such as connectomic “fingerprinting” (Shen et al., 2017) may reveal wider functional interactions and topological configurations that predict individual differences in long-term memory retrieval.”

The second experiment, for me, would be greatly strengthened by more contextualization for the ROIs studied in the Introduction (although at this stage, that would of course itself be posthoc – but that can't be helped), and establishing a control analysis and/or scenario in which there would be an alternative outcome. For example, the authors note that some brain areas have been attributed distinctly to recollection vs semantic processing – if they were to show that these areas, in contrast to IFG have memory domain-specific connectivity profiles, or that in contrast to IFG-vmPFC there are IFG-XXX patterns that distinguish the memory types, this would tell us something about the specificity of these effects. Is IFG a general control region that is agnostic to memory type, in contrast to ___? Or perhaps it is a hub that is involved in control for different forms of memory, but it does differentially engage, as needed, with other cortical nodes that discriminate these forms of memory.

We sincerely appreciate the insightful suggestions made by the reviewer to further improve our results in Experiment 2. In order to further address the specificity of the observed results to memory retrieval, we employed two sets of additional measurements that were collected from this participant cohort. While fluid intelligence was measured via proportion of correct trials out of the number attempted trials in Raven's Advanced Progressive Matrices (Raven and Raven, 2003), reaction time to incongruent trials in the Flanker task was used to assess selective attention / inhibitory control (Eriksen and Eriksen, 1974). Neither individual differences in fluid intelligence (partial $r_p = -.010$, $p = .45$), nor selective attention / inhibitory control (partial $r_p = .052$, $p = .27$) showed a significant correlation with individual variation across the functional connectivity between LIFG/aINS and vmPFC (corrected for age, gender and in-scanner head motion). We now include these findings in the manuscript and the supplementary materials. Specifically, we changed the relevant results section as follows:

“Moreover, individual variation in the connectivity strength of this neural link was neither related to fluid intelligence (partial $r_p = -.010$, $p = .45$), nor selective attention / inhibitory control (partial $r_p = .052$, $p = .27$) as measured via accuracy-based indices in the Raven's advanced progressive matrices (Raven and Raven, 2003) and reaction time to incongruent

trials in the Flanker (Eriksen and Eriksen, 1974) tasks, respectively (Supplementary Information S2).”

Figure S2. Differential neural circuits related to long-term memory retrieval. Two separate linear regressions were carried out in order to investigate neural circuits that were differentially associated with performance differences across the retrieval of weakly associated semantic and episodic memory. Across participants, (A) while *reduced positive connectivity (or greater anti-correlation)* of the LIFG/aINS cluster to the bilateral posterior cingulate/retrosplenial cortices (PCC/RSC) was associated with better performance (lower inverse efficiency score) in the retrieval of weak semantic memories, (B) *reduced* connectivity of the same seed region to the ventromedial prefrontal cortex (vmPFC) was linked to selective advantage in the retrieval of weak episodic memories. Additional seed-based connectivity of the PCC/RSC and vmPFC clusters identified largely overlapping connectivity profiles that were centered on the default mode network. In addition, the meta-analytic decoding of the LIFG/aINS – PCC/RSC and LIFG/aINS – vmPFC revealed comparable terms that spanned “recollection, semantic, episodic and autobiographical memory” as well as distinct terms such as “navigation and incentive delay”. Straight lines represent the best fit, while shaded areas illustrate 95% confidence intervals.

Furthermore, following the reviewer’s suggestions, we aimed to assess differential links of the identified LIFG/aINS cluster in relation to semantic versus episodic memory performance in this cohort. The results revealed that while reduced positive connectivity (or greater anti-correlation) between the LIFG/aINS to the vmPFC was associated with better performance in the episodic memory task, reduced connectivity of the same LIFG/aINS seed to the posterior cingulate (PCC) and retrosplenial cortices (RSC) was associated with better performance in the semantic memory task. Taken together, these results suggest a common neural circuitry centered on the LIFG/aINS and vmPFC connectivity, but with greater need for separating LIFG/aINS and PCC for selective advantage in the performance of a semantic than episodic memory task. Both PCC/RSC and the vmPFC are central hubs of the default mode network. As such, this collective evidence indicates the need to separate fronto-parietal and default mode networks, to control memory traces in order to perform well under conditions of high demand. Nevertheless, further research will be required to deduce the causal influence of the identified region in the controlled retrieval of memory across participants. We have now added this information to the supplementary materials and changed the results section as follows:

“In addition to this shared neural circuitry, further analysis also revealed stronger impact of the anti-correlation between the employed seed region and retrosplenial, posterior cingulate cortices on semantic than episodic memory retrieval (Fig. S6).”

Additionally, the authors put a fair bit of emphasis on the neural correlates being for “weakly associated” memory traces – but this doesn’t appear to be well-characterized in their individual differences analysis. By looking at the contrast of weak vs strong, the reader is left to wonder where the “movement” in the data lie across people. Could the authors unpack this a bit by characterizing different connectivity profiles in Experiment 2 according to strong vs weak performance? For example, some people presumably struggle more with both strong and weak conditions, some more with weak specifically, and some more for semantic vs episodic. I believe there’s an opportunity to learn more about what aspects of individual memory performance are tracked by IFG resting-state connectivity.

We thank the reviewer for these extensive suggestions. To reiterate, the aim of our second experiment was to investigate neural links with the LIFG/aINS cluster that would relate to selective advantage in the retrieval of weakly encoded information across both semantic and episodic memory. Although, we fully agree that further investigations might reveal interesting aspects of neural interactions in this context, this was beyond the scope of our current study, it will be exciting and necessary to test these hypotheses in future studies.

References

- Badre D, Wagner AD (2007) Left ventrolateral prefrontal cortex and the cognitive control of memory. *Neuropsychologia* 45:2883-2901.
- Barredo J, Oztekin I, Badre D (2015) Ventral fronto-temporal pathway supporting cognitive control of episodic memory retrieval. *Cereb Cortex* 25:1004-1019.
- Binder JR, Desai RH, Graves WW, Conant LL (2009) Where is the semantic system? A critical review and meta-analysis of 120 functional neuroimaging studies. *Cereb Cortex* 19:2767-2796.
- Davey J, Cornelissen PL, Thompson HE, Sonkusare S, Hallam G, Smallwood J, Jefferies E (2015) Automatic and Controlled Semantic Retrieval: TMS Reveals Distinct Contributions of Posterior Middle Temporal Gyrus and Angular Gyrus. *J Neurosci* 35:15230-15239.
- Eriksen BA, Eriksen CW (1974) Effects of noise letters upon the identification of a target letter in a nonsearch task. *Percept Psychophys* 16:143-149.
- Finn ES, Shen X, Scheinost D, Rosenberg MD, Huang J, Chun MM, Papademetris X, Constable RT (2015) Functional connectome fingerprinting: identifying individuals using patterns of brain connectivity. *Nat Neurosci* 18:1664-1671.
- Gilboa A, Marlatte H (2017) Neurobiology of Schemas and Schema-Mediated Memory. *Trends Cogn Sci* 21:618-631.
- Irish M, Vatansever D (2020) Rethinking the episodic-semantic distinction from a gradient perspective. *Current Opinion in Behavioral Sciences* 32:43-49.
- Jackson RL (2020) The neural correlates of semantic control revisited. *Neuroimage* 224:117444.
- Jefferies E (2013) The neural basis of semantic cognition: converging evidence from neuropsychology, neuroimaging and TMS. *Cortex* 49:611-625.
- Noonan KA, Jefferies E, Visser M, Lambon Ralph MA (2013) Going beyond inferior prefrontal involvement in semantic control: evidence for the additional contribution of dorsal angular gyrus and posterior middle temporal cortex. *J Cogn Neurosci* 25:1824-1850.
- Raven J, Raven J (2003) Raven Progressive Matrices. In: *Handbook of nonverbal assessment.*, pp 223-237. New York, NY, US: Kluwer Academic/Plenum Publishers.
- Renoult L, Irish M, Moscovitch M, Rugg MD (2019) From Knowing to Remembering: The Semantic-Episodic Distinction. *Trends Cogn Sci* 23:1041-1057.
- Ritchey M, Cooper RA (2020) Deconstructing the Posterior Medial Episodic Network. *Trends Cogn Sci* 24:451-465.
- Shen X, Finn ES, Scheinost D, Rosenberg MD, Chun MM, Papademetris X, Constable RT (2017) Using connectome-based predictive modeling to predict individual behavior from brain connectivity. *Nat Protoc* 12:506-518.
- Thakral PP, Madore KP, Schacter DL (2017) A Role for the Left Angular Gyrus in Episodic Simulation and Memory. *J Neurosci* 37:8142-8149.
- Tulving E, Donaldson W, Bower GH, United States. Office of Naval Research. (1972) *Organization of memory*. New York: Academic Press.
- Vatansever D, Menon DK, Stamatakis EA (2017a) Default mode contributions to automated information processing. *Proc Natl Acad Sci U S A* 114:12821-12826.
- Vatansever D, Bzdok D, Wang HT, Mollo G, Sormaz M, Murphy C, Karapanagiotidis T, Smallwood J, Jefferies E (2017b) Varieties of semantic cognition revealed through simultaneous decomposition of intrinsic brain connectivity and behaviour. *Neuroimage* 158:1-11.
- Whitney C, Kirk M, O'Sullivan J, Lambon Ralph MA, Jefferies E (2011) The neural organization of semantic control: TMS evidence for a distributed network in left inferior frontal and posterior middle temporal gyrus. *Cereb Cortex* 21:1066-1075.

- Whitney C, Kirk M, O'Sullivan J, Lambon Ralph MA, Jefferies E (2012) Executive semantic processing is underpinned by a large-scale neural network: revealing the contribution of left prefrontal, posterior temporal, and parietal cortex to controlled retrieval and selection using TMS. *J Cogn Neurosci* 24:133-147.
- Yeo BT, Krienen FM, Sepulcre J, Sabuncu MR, Lashkari D, Hollinshead M, Roffman JL, Smoller JW, Zollei L, Polimeni JR, Fischl B, Liu H, Buckner RL (2011) The organization of the human cerebral cortex estimated by intrinsic functional connectivity. *J Neurophysiol* 106:1125-1165.

Reviewer #1 (Remarks to the Author):

The authors have appropriately revised their manuscript and clarified a number of issues, as well as addressed limitations at the end of the discussion.

Line 387: Did the authors meant to say pattern completion? (would fit better in this context and the focus of the paper on retrieval)

Reviewer #2 (Remarks to the Author):

The authors have attentively addressed all of my comments and concerns (as well as those raised by the other reviewers), and the manuscript has been strengthened as a result. I appreciate the thoughtfulness that went into these revisions, and I continue to believe that this paper will be read with great interest by many in our field.

I have only one lingering quibble. In their response letter, the authors claim that they have included a "circularity warning" for Fig 4A to indicate that the scatterplots are shown for illustrative purposes only, and yet I cannot see any text to that effect in the figure caption. Inferential circularity continues to plague our field, and many readers can not readily distinguish which aspects of a figure are depicting actual data and which are simply an illustration of the region-defining contrast/correlation (the latter inherently inflate the apparent effect size). I would like the authors to remedy this prior to publication by amending the caption, as they have already done for Fig 3C-D.

Minor typo:

p. 12, line 507-508: "The stimuli.. was visually presented" should read "The stimuli... were visually presented"

Reviewer #3 (Remarks to the Author):

Overall, I found the authors' revision thorough and responsive. In particular, the revisions to the framing of the study in the Introduction and to the specific methods/results has been greatly strengthened.

I believe this will make a thought-provoking contribution to the field

Reviewer #1 (Remarks to the Author):

The authors have appropriately revised their manuscript and clarified a number of issues, as well as addressed limitations at the end of the discussion.

Line 387: Did the authors meant to say pattern completion? (would fit better in this context and the focus of the paper on retrieval)

We thank the reviewer for their positive assessment of our manuscript and revision. In line with the reviewer's request, we have now altered the sentence on Line 387 to read "pattern completion" instead of pattern separation.

Reviewer #2 (Remarks to the Author):

The authors have attentively addressed all of my comments and concerns (as well as those raised by the other reviewers), and the manuscript has been strengthened as a result. I appreciate the thoughtfulness that went into these revisions, and I continue to believe that this paper will be read with great interest by many in our field.

I have only one lingering quibble. In their response letter, the authors claim that they have included a "circularity warning" for Fig 4A to indicate that the scatterplots are shown for illustrative purposes only, and yet I cannot see any text to that effect in the figure caption. Inferential circularity continues to plague our field, and many readers can not readily distinguish which aspects of a figure are depicting actual data and which are simply an illustration of the region-defining contrast/correlation (the latter inherently inflate the apparent effect size). I would like the authors to remedy this prior to publication by amending the caption, as they have already done for Fig 3C-D.

Minor typo:

p. 12, line 507-508: "The stimuli.. was visually presented" should read "The stimuli... were visually presented".

We thank the reviewer for their thorough and insightful comments and the positive assessment of our manuscript. We apologise for the lack of clarity on our Figure 4 and have now altered the caption in line with the reviewer's request. Specifically, we added the following line: *"The scatter plot is shown for illustrative purposes only. While the straight lines represent the best linear fit over individual values, shaded areas illustrate 95% confidence intervals."* In addition, we corrected the typo on Lines 507-508.

Reviewer #3 (Remarks to the Author):

Overall, I found the authors' revision thorough and responsive. In particular, the revisions to the framing of the study in the Introduction and to the specific methods/results has been greatly strengthened.

I believe this will make a thought-provoking contribution to the field.

We thank the reviewer for their positive assessment of our study and the corresponding manuscript.